# Virtual intracranial EEG signals reconstructed from MEG with potential for epilepsy surgery

Miao Cao[1,2,13], Daniel Galvis [3,4,5,6,13], Simon J. Vogrin[1,2,7], William P. Woods[7], Sara Vogrin[1,8], Fan Wang[9,10,11], Wessel Woldman[3,4,5,6], John R. Terry[3,4,5,6], Andre Peterson[1,2,12], Chris Plummer [1,2,7✉] & Mark J. Cook[1,2,12]

Modelling the interactions that arise from neural dynamics in seizure genesis is challenging but important in the effort to improve the success of epilepsy surgery. Dynamical network models developed from physiological evidence offer insights into rapidly evolving brain networks in the epileptic seizure. A limitation of previous studies in this field is the dependence on invasive cortical recordings with constrained spatial sampling of brain regions that might be involved in seizure dynamics. Here, we propose virtual intracranial electroencephalography (ViEEG), which combines non-invasive ictal magnetoencephalographic imaging (MEG), dynamical network models and a virtual resection technique. In this proof-of-concept study, we show that ViEEG signals reconstructed from MEG alone preserve critical temporospatial characteristics for dynamical approaches to identify brain areas involved in seizure generation. We show the non-invasive ViEEG approach may have some advantage over intracranial electroencephalography (iEEG). Future work may be designed to test the potential of the virtual iEEG approach for use in surgical management of epilepsy.

[1] Department of Medicine St Vincent's Hospital, The University of Melbourne, Melbourne, Australia. [2] Centre for Clinical Neurosciences and Neurological Research, St Vincent's Hospital Melbourne, Melbourne, Australia. [3] Translational Research Exchange at Exeter, University of Exeter, Exeter, UK. [4] Living Systems Institute, University of Exeter, Exeter, UK. [5] Centre for Systems Modelling and Quantitative Biomedicine, University of Birmingham, Birmingham, UK. [6] Institute of Metabolism and Systems Research, University of Birmingham, Birmingham, UK. [7] Faculty of Health, Art and Design, Swinburne University of Technology, Melbourne, Australia. [8] Department of Medicine Western Health, The University of Melbourne, Melbourne, Australia. [9] State Key Laboratory of Brain and Cognitive Science, Institute of Biophysics, Chinese Academy of Sciences, Beijing, China. [10] CAS Centre for Excellence in Brain Science and Intelligence Technology, Beijing, China. [11] University of Chinese Academy of Sciences, Beijing, China. [12] Department of Biomedical Engineering, The University of Melbourne, Melbourne, Australia. [13] These authors contributed equally: Miao Cao, Daniel Galvis. ✉email: chris.plummer@svha.org.au

Modelling complex systems like the brain is challenging[1], particularly when studying time-evolving interactions that arise from normal and aberrant neural dynamics across multiple temporal and spatial scales[2–4]. Patterns of brain activation and interactions form the neural correlates of brain states and behaviour, which are fundamental to understanding the underlying mechanisms of brain function[5]. Developing and validating mathematical and computational models of brain function and neural dynamics has been a key mission for neuroscience research in the last few decades.

Dynamical network models provide great capacity to probe the underlying mechanisms of complex neural dynamics. Inspired by early studies of excitatory and inhibitory neurons, investigators have developed dynamical models of neural mass and neural mass networks, which connect an ensemble of neural mass models into macroscopic neural systems[1,6–9]. Employing dynamical models, multiple attempts have been made to understand the mechanisms underlying normal[10,11] and pathological neural dynamics[12–14]. Dynamical network models have also been applied to neurophysiological data recorded from the human brain to develop specific hypotheses towards clinical application[15–18].

A major limitation of previous studies in this field is the dependence on invasive intracranial electrode recordings of cortical activity when applying dynamical network models to experimental data. This can lead to insufficient sampling of brain networks with the biased representation of the actual systems involved[19]. Moreover, invasive cortical recordings are relatively costly to obtain from animal models and only possible from the diseased human brain, as in epilepsy surgery planning, with inherent risks[20,21]. Non-invasive neuroimaging, such as high-density encephalography (HDEEG) and magnetoencephalography (MEG), offers high temporal resolution and whole-brain spatial coverage[22–26]. Unlike functional magnetic resonance imaging (fMRI), electromagnetic fields recorded over the head surface by EEG and MEG represent the linear summation of collective source activity[24,26,27]. Advances in EEG and MEG source imaging techniques have improved the capacity to project the recorded surface electromagnetic fields back to source activity with high spatial resolution when sufficient signal-to-noise ratios are obtained[27–29].

Epilepsy affects about 1% of the global population, and at least one-third of epilepsy patients have seizures that are refractory to medication[30,31]. While surgery can be an effective treatment for pharmaco-refractory epilepsy, it is widely underused[32]. Presurgical evaluation is particularly challenging when MRI shows no clear lesion, a large complex lesion, or multiple potential epileptogenic lesions[33,34]. Such cases stand to benefit from non-invasive dynamical approaches to better characterise the brain networks involved in seizure dynamics. Here, we aim to apply a dynamical network approach to source reconstructed non-invasive ictal MEG data, and we introduce the concept of virtual intracranial EEG (ViEEG). A set of sources, or virtual electrodes[35], are defined in the reconstructed individualised MRI brain where ictal source signals can be reconstructed as ViEEG signals. We then construct patient-specific functional networks integrated into dynamical network models to determine the effects of the network structure on seizure transitions. Specifically, brain regions responsible for seizure generation are identified by evaluating the contribution of each node to the network excitability, i.e., the likelihood of a particular node affecting the transition to a seizure. From this evaluation, we attempt to define a virtual ictogenic zone (VIZ) non-invasively and assess our ViEEG derived dynamical network models against the preoperative HDEEG and MEG ictal source localisation (ESL, MSL), the clinical intracranial EEG (iEEG) localisation, and the putative epileptogenic zone (EZ) as defined by resection linked to long-term postoperative outcome. In doing this, we aim to address several questions. Can we reconstruct ictal ViEEG signals with distinct spatial and temporal characteristics of epileptiform discharges? If so, can dynamical network models using ictal ViEEG signals guide a surgical strategy that characterises the EZ while identifying brain areas that are less likely to be involved in the EZ? Finally, can dynamical network models using ictal ViEEG signals provide unique information to the current clinical localisation, including the iEEG seizure onset zone (SOZ) and HDEEG and MEG source localisation, in characterising the EZ?

This study uses a multi-disciplinary approach to objectively characterise the ictal brain networks from non-invasive MEG data without the risks and constraints of iEEG. We demonstrate that non-invasive ictal ViEEG signals contain meaningful temporo-spatial data to assist the characterisation of the putative EZ. We apply dynamical network models and a virtual resection technique to ictal ViEEG signals to identify a sub-network VIZ that helps elucidate the EZ (Fig. 1). We find that alternative surgical strategies can be devised from VIZ results for non-seizure-free patients. More crucially, we show that the VIZ from MEG data alone can predict the earliest solution out of MSL and ESL. The earliest solution best informs the likely EZ in our recent study using simultaneous HDEEG and MEG data[36].

## Results

Twelve patients with 25 seizures captured by simultaneously acquired HDEEG and MEG were included in our analysis (Table 1, Supplementary Table 6).

**Clinical features of ictal ViEEG signals**. The first question we addressed is whether we can reconstruct ictal ViEEG signals with distinct clinical characteristics to enable dynamical network models to identify brain areas that are involved and not involved in increased in silico seizure likelihood. In Fig. 2, we demonstrate a side-by-side comparison of ViEEG and iEEG signals from Patient 5. Ictal waveforms from ViEEG can be visually identified solely from sources in anterior hippocampal and basal temporal structures, which is consistent with the iEEG SOZ. Representative ictal ViEEG signals of 25 seizures are shown in Supplementary Figs. 5–16. Further examples of ictal ViEEG against iEEG are given in Supplementary Fig. 17.

**VIZ hotspot and boundary characterise the EZ and predict the clinical localisation**. The next question we asked is whether the VIZ can devise a strategy that characterises the EZ with sufficient precision and recall. Successful prediction of the likely EZ by the VIZ hotspot is demonstrated in Fig. 2. Although the VIZ boundary is much more extensive than the proposed EZ, it captures its entirety and also identifies non-ictogenic brain areas. The dynamical network models (using two different network inference techniques, AEC-VIZ and MI-VIZ) obtain high recall in predicting the EZ (Fig. 3B for seizure-free group). Mixed-effects logistic regression models suggest AEC-VIZ and MI-VIZ significantly predict the resection margin at nodal level (Fig. 3A). Figure 3B shows the precision, or positive predictive value, of the VIZ hotspot (top 20% VIZ nodes ranked by NI) and the recall, or sensitivity, of the VIZ boundary in predicting the resection margin and the earliest solution. The MI-VIZ recall (for VIZ boundary) sufficiently captures the entirety of resection margin and the earliest solution and identifies non-ictogenic brain areas that are less likely to overlap with the EZ and are therefore potentially less concerning for iEEG coverage. Moderate precision values (for VIZ hotspot) are found for both AEC-VIZ and MI-VIZ in predicting the resection margin and earliest solution. MI-

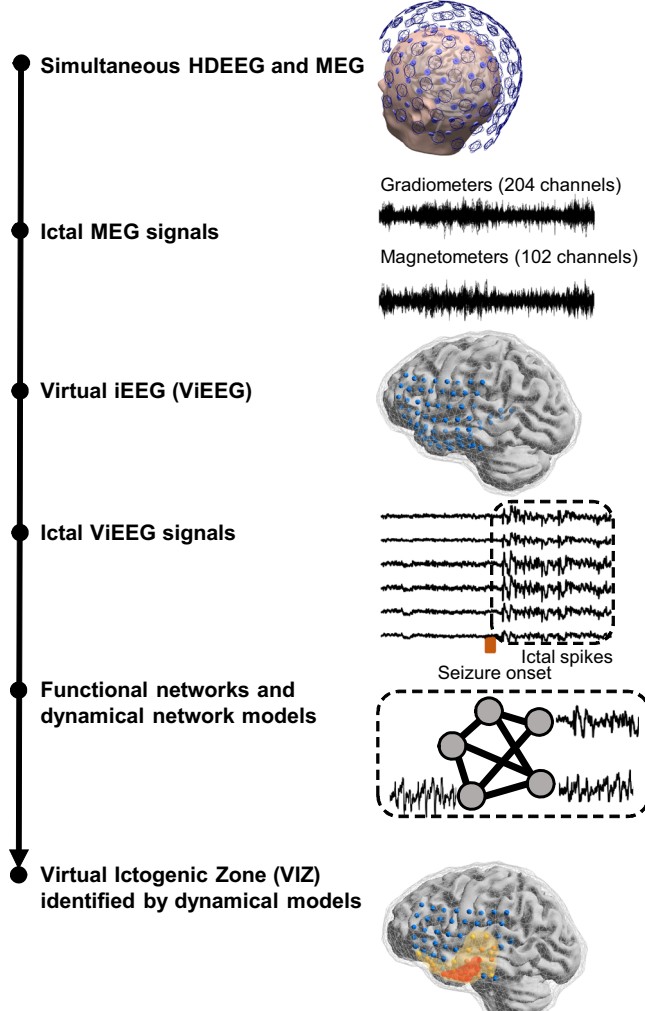

- **Simultaneous HDEEG and MEG**
- **Ictal MEG signals**

  Gradiometers (204 channels)

  Magnetometers (102 channels)
- **Virtual iEEG (ViEEG)**
- **Ictal ViEEG signals**

  Seizure onset    Ictal spikes
- **Functional networks and dynamical network models**
- **Virtual Ictogenic Zone (VIZ) identified by dynamical models**

**Fig. 1 Workflow of Virtual iEEG (ViEEG) and network model.**
Simultaneous HDEEG and MEG were acquired from surgical candidates in presurgical evaluation for epilepsy surgery[36]. Ictal MEG signals from 102-channel magnetometers and 204-channel gradiometers are epoched and pre-processed for source signal reconstruction. ViEEG locations fully contain MSL solutions (early, mid and late phases of averaged ictal discharges)[36] and the entire resection margin. Ictal ViEEG signals are reconstructed using a beamformer technique and a boundary element method (BEM) model generated from individual MRI scans. Functional networks are constructed using two connectivity methods, amplitude envelope correlation (AEC) and mutual information (MI), and dynamical network models are applied to evaluate how cortical excitability changes when a node is virtually removed from the network. The Virtual Ictogenic Zone (VIZ), identified by a dynamical network approach, consists of nodes that decrease cortical excitability when virtually resected from the network. We hypothesise that this VIZ helps elucidate the epileptogenic zone (EZ) and identifies non-ictogenic brain areas that are less likely to be involved in the EZ.

VIZ hotspots appear to have higher precision than AEC-VIZ hotspots in predicting the resection margin and, to a lesser degree, the earliest solution. The corresponding *F*-scores (harmonic mean of precision of VIZ hotspot and recall of VIZ boundary) for MI-VIZ (median = 0.75) in predicting the EZ (i.e., resection margin for seizure-free patients) suggest our approach helps delineate the EZ and identify non-ictogenic brain regions based on recall of MI-VIZ (Supplementary Fig. 4). The spatial overlap between the VIZ and clinical localisation are

**Table 1 Patient information, MEG seizures, earliest source localisation solution and outcome.**

| Patient | MRI | Histology | MEG seizures recorded | MEG seizures reconstructed | Earliest solution from ESL and MSL | Engel outcome |
|---|---|---|---|---|---|---|
| 1 | Normal | CD 1A | 6 | 1 | MSL | I (Rare non-disabling seizures) |
| 2 | Normal | CD 1 | 2 | 2 | ESL | I (Seizure free) |
| 3 | Normal | CD 2A | 7 | 6 | MSL | I (Rare non-disabling seizures) |
| 4 | Normal | Non-specific | 8 | 5 | MSL | III (Fewer disabling seizures) |
| 5 | Normal | CD 1C | 2 | 1 | MSL | I (Seizure free) |
| 6 | Normal | CD 1 | 1 | 1 | ESL | I (Seizure free) |
| 7 | Normal | CD 2A | 2 | 1 | Non-localising | I (Seizure free) |
| 8 | Normal | CD 1 | 2 | 2 | ESL | I (Seizure free) |
| 9 | Normal | CD 2B | 2 | 1 | ESL | II (Rare disabling seizures) |
| 10 | Multi-lobar dysplasia | Normal | 1 | 1 | MSL | I (Non-disabling seizures) |
| 11 | Multi-lobar dysplasia | Ischaemia | 2 | 2 | MSL | III (Fewer disabling seizures) |
| 12 | Right frontal gliosis | Gliosis | 2 | 2 | ESL | I (Seizure free) |

Twelve patients had 36 seizures recorded by MEG. 25/36 seizures reconstructed using ViEEG present distinct morphological features of epileptiform discharges in source space. Six patients achieved seizure-freedom. Nine patients had complex lesions. Early-MSL predicted the EZ in six patients (Patients 1, 3, 4, 5, 10, 11), while early-ESL predicted the EZ in five patients (Patients 2, 6, 8, 9, 12). Ictal discharges were not localisable in one patient (Patient 7). One patient from the original 13 patients published by Plummer et al[36] with normal MRI only had interictal discharges captured by HDEEG and MEG and hence was not included in the present study. CD cortical dysplasia, HDEEG high density EEG, ESL HDEEG source localisation, MSL MEG source localisation, ViEEG virtual intracranial EEG, EZ epileptogenic zone.

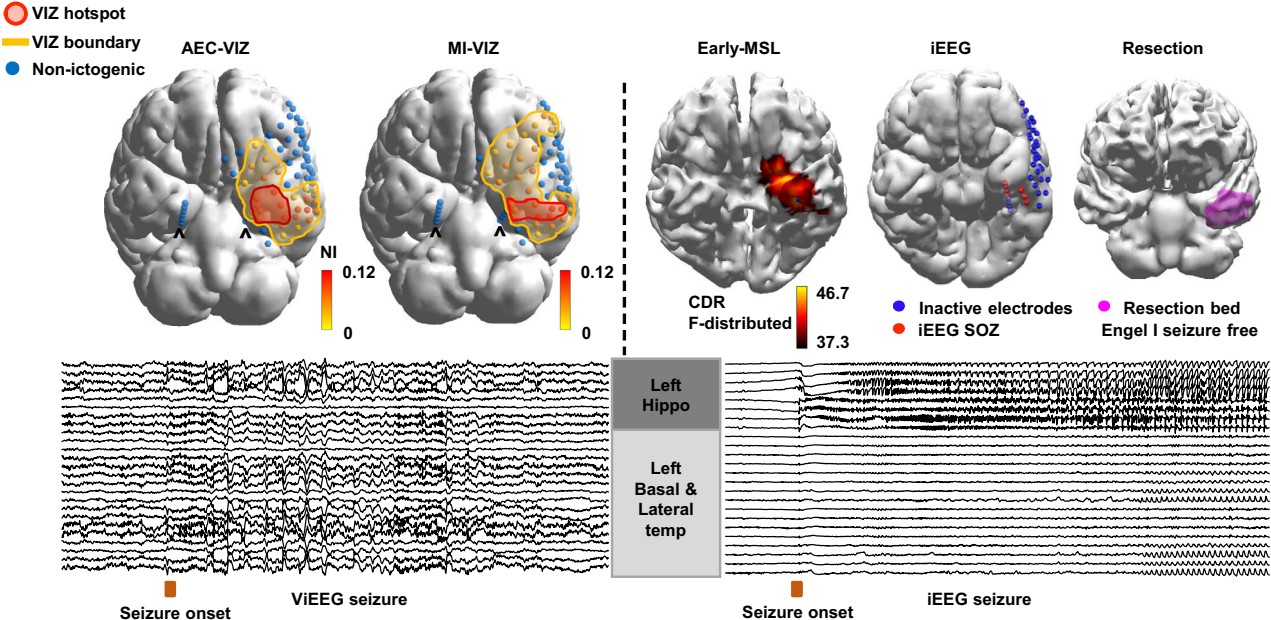

**Fig. 2 (Patient 5) Morphological and spatial characteristics of ictal ViEEG signals and concordance between VIZ, iEEG SOZ, and early-MSL.** While carrying a slightly different morphology to the corresponding iEEG ictal rhythm, distinct ictal waveforms are seen in the left anterior hippocampal structure and left basal temporal region from a ViEEG seizure aligned in time with seizure onset identified by MEG sensor signals. This is spatially concordant with a seizure captured by iEEG after the MEG recording, where the seizure starts from the left anterior hippocampus and spreads to the left basal temporal area. AEC-VIZ and MI-VIZ identified by dynamical network models are similar with boundaries containing the iEEG SOZ, resection margin and early-MSL (earliest ictal source localisation solution accounting for 90% of the signal variance from averaged discharge take-off towards the peak)[31]. Both AEC-VIZ and MI-VIZ hotspots (high NI values) point to the left anterior hippocampus, entorhinal cortex and basal temporal structures. The patient has been seizure-free for over 2 years post-surgery and histology showed cortical dysplasia at the entorhinal cortex. Both AEC-VIZ and MI-VIZ appear to have captured the likely EZ (ViEEG hippocampal depth electrodes are denoted by hat symbols to distinguish them from the left lateral temporal ViEEG grid electrodes). *Abbreviations*: MEG magnetoencephalography, iEEG intracranial electroencephalography, ViEEG virtual intracranial electroencephalography, EZ epileptogenic zone, SOZ seizure onset zone, HDEEG high density electroencephalography, VIZ virtual ictogenic zone, MSL MEG source localisation, ESL HDEEG source localisation, AEC amplitude envelope correlation, MI mutual information, AEC-VIZ virtual ictogenic zone using amplitude envelope correlation, MI-VIZ virtual ictogenic zone using mutual information, NI node ictogenicity.

demonstrated on a per patient and per seizure basis in Supplementary Tables 4 and 5. Furthermore, we find AEC-VIZ and MI-VIZ hotspots and boundaries predict the iEEG SOZ and the early-MSL solution (Fig. 3A, Supplementary Tables 2 and 3). AEC-VIZ and MI-VIZ hotspots and boundaries are less likely to predict mid-MSL and late-MSL solutions (Fig. 3A, Supplementary Tables 2 and 3), particularly when mid-MSL and late-MSL do not overlap with the corresponding early-MSL solution— Patient 2 (Supplementary Fig. 6), Patient 6 (Supplementary Fig 10), Patient 11 (Fig. 4, Supplementary Fig 15), Patient 12 (Supplementary Fig. 16).

**MI-VIZ predicts the optimal source localisation solution.** The earliest solution (the first-occurring early-MSL or early-ESL sLORETA solution) has been reported as the best predictor of the EZ[36]. In this study, we are interested in whether VIZ identified by dynamical network models predicts the earliest solution and not just the early-MSL. AEC-VIZ and MI-VIZ significantly predict the earliest solution (Fig. 3A, Supplementary Tables 2 and 3). Specifically, we find that the MI-VIZ predicts the early-ESL instead of early-MSL in three patients—Patient 6 (Fig. 5, Supplementary Fig. 10), Patient 8 (Supplementary Fig. 12), and Patient 12 seizure 1 (Fig. 6, Supplementary Fig. 16)—out of the total five patients who have early-ESL as the earliest solution. This is also demonstrated in Fig. 3A as higher ORs are seen for MI-VIZ hotspot and boundary prediction of the earliest solution against the early-MSL alone (also see Supplementary Tables 2 and 3).

**MI-VIZ may predict the putative EZ and clinical localisation better than AEC-VIZ.** AIC and BIC were calculated from mixed-effects logistic regression models evaluating a statistical association between AEC-VIZ and MI-VIZ and the resection margin as well as the clinical localisation (Supplementary Tables 2 and 3). Lower AIC and BIC values suggest higher predictive power for the model. As shown in Supplementary Table 3, the MI-VIZ boundary consistently has lower AIC and BIC values than the AEC-VIZ boundary in predicting the resection margin and clinical localisation. Further, the MI-VIZ hotspot has lower AIC and BIC values than corresponding AEC-VIZ hotspot values in predicting the resection, the earliest solution, early-MSL and late-MSL solutions, while AEC-VIZ hotspots better predict the iEEG SOZ and mid-MSL solution than MI-VIZ hotspots (Supplementary Table 2). Examples can be seen in Patient 6 (Fig. 5, Supplementary Fig. 10) where the MI-VIZ hotspot predicts the successful repeat surgery resection bed while the more diffuse AEC-VIZ hotspot encompasses the first failed resection bed as well and in Patient 12 seizure 1 (Fig. 6, Supplementary Fig. 16), where the MI-VIZ hotspot predicts the resection margin and the earliest solution, while AEC-VIZ hotspot does not.

**Discussion**

This study reconstructs ictal source signals using a high number of spontaneous seizures captured by MEG[36]. Ictal ViEEG signals from at least one seizure per patient present distinct characteristics of ictal events, such as hyper-synchronised rhythms, clear transitions from background activity to a seizure state, and spatial

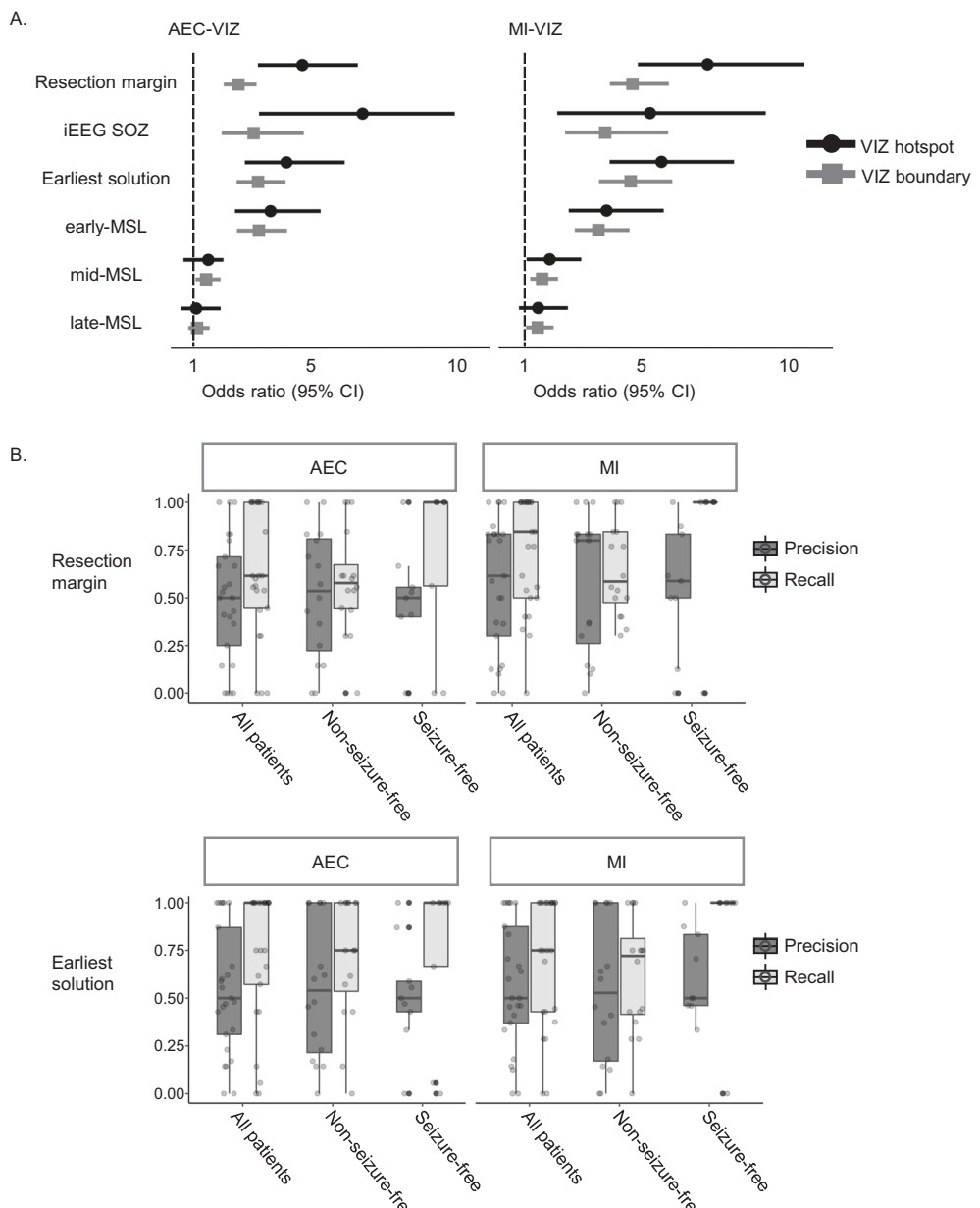

**Fig. 3 Statistical findings of AEC and MI-VIZ predicting the resection margin and clinical localisation. A** Odds ratios (ORs) with 95% confidence intervals are presented for VIZ hotspot and boundary predicting the resection margin, iEEG SOZ, the earliest solution and MSL solutions (early, mid and late). ORs for AEC-VIZ and MI-VIZ hotspots are consistently higher than those of AEC-VIZ and MI-VIZ boundary in predicting the resection margin, iEEG SOZ and the earliest solution. MI-VIZ (hotspot and boundary) have higher ORs than the corresponding AEC-VIZ results when predicting the resection margin. As opposed to AEC-VIZ, MI-VIZ is more likely to predict the resection margin than iEEG SOZ. The upper and lower boundaries of error bars are defined as upper and lower boundaries of 95% confidence intervals. **B** Precision (or positive predictive value) and recall (or sensitivity) for AEC-VIZ and MI-VIZ in predicting the resection margin (top panel) and the earliest solution (bottom panel) are presented in boxplots (horizontal bar, box upper boundary, box lower boundary and dots represent median, first and third quartile and each individual VIZ, respectively). This shows that the MI-VIZ recall (for VIZ boundary) sufficiently captures the entirety of resection margin and the earliest solution and identifies non-ictogenic brain areas that are less likely to overlap with the EZ and are therefore potentially less concerning for iEEG coverage. Moderate precision values (for VIZ hotspot) are found for both AEC-VIZ and MI-VIZ in predicting the resection margin and earliest solution. MI-VIZ hotspots appear to have higher precision than AEC-VIZ hotspots in predicting the resection margin and, to a lesser degree, the earliest solution. The spatial overlap between VIZ and clinical localisation are demonstrated on a per patient and per seizure basis in Supplementary Table 4. Note that odds ratios, precision, and recall results are based on $n = 25$ seizures with $n = 9$ seizures (from seizure-free group of 6 patients) and $n = 16$ seizures (from non-seizure-free group of 6 patients). Note that box plot minima = minimum value in the data, maxima = maximum value in the data, centre = median, upper boundary of box is 75th percentile, lower boundary of box is 25th percentile, lower boundary of whisker is defined as 25th percentile minus 1.5 times interquartile range ($Q_3-Q_1$), i.e. $Q_1-1.5*(Q_3-Q_1)$ and upper boundary of whisker is defined as 75th percentile plus 1.5 times interquartile range ($Q_3-Q_1$), i.e. $Q_3 + 1.5*(Q_3-Q_1)$. Black dot indicates outlier. *Abbreviations*: MEG magnetoencephalography, iEEG intracranial electroencephalography, ViEEG virtual intracranial electroencephalography, EZ epileptogenic zone, SOZ seizure onset zone, HDEEG high density electroencephalography, VIZ virtual ictogenic zone, MSL MEG source localisation, ESL HDEEG source localisation, AEC amplitude envelope correlation, MI mutual information, AEC-VIZ virtual ictogenic zone using amplitude envelope correlation, MI-VIZ virtual ictogenic zone using mutual information.

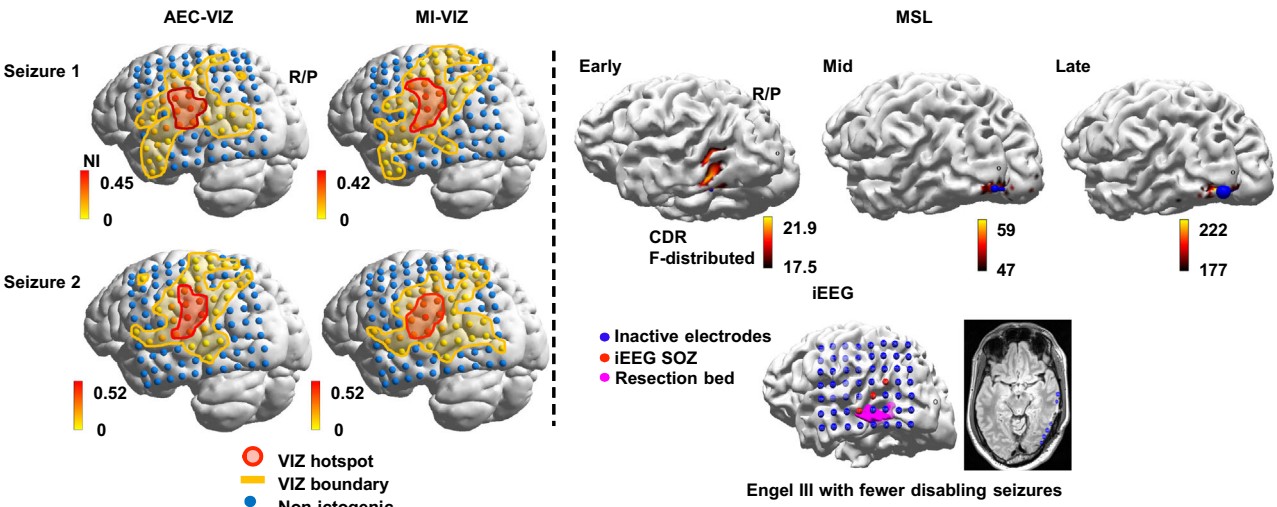

**Fig. 4 (Patient 11) AEC-VIZ and MI-VIZ hotspot concordance with the earliest source localisation solution (early-MSL) and resection margin across two seizures suggests an alternative surgical strategy.** An alternative surgical plan is suggested by AEC-VIZ and MI-VIZ hotspots (left posterior temporal–parietal junction) from both seizures as opposed to the more discrete left tempo-parieto-occipital junction localisation given by the early-MSL solution. Both VIZ solutions overlapped the iEEG SOZ but lie outside the resection bed (magenta). The patient had a suboptimal surgical outcome (Engel III with fewer disabling seizures). It is possible that the concordant VIZ hotspots and the iEEG flag a more extensive EZ or a separate EZ network. *Abbreviations*: MEG magnetoencephalography, iEEG intracranial electroencephalography, ViEEG virtual intracranial electroencephalography, EZ epileptogenic zone, SOZ seizure onset zone, HDEEG high density electroencephalography, VIZ virtual ictogenic zone, MSL MEG source localisation, ESL HDEEG source localisation, AEC amplitude envelope correlation, MI mutual information, AEC-VIZ virtual ictogenic zone using amplitude envelope correlation, MI-VIZ virtual ictogenic zone using mutual information, NI node ictogenicity.

patterns of seizure propagation. Such qualitative characteristics of ictal ViEEG are also reflected by corresponding ictal iEEG data from the 6 patients whose iEEG data were available to us (Fig. 2, Supplementary Fig. 17). At this point, however, we lack a reliable quantitative measure to define the relationship between ViEEG reconstructed signals and the corresponding iEEG discharges for morphology, spatial topography, and temporal evolution on an individual patient level. While requiring further investigation, this proof of concept work does, nonetheless, suggest that our ViEEG-derived VIZ does have the potential to serve as a useful biomarker for the patient's putative EZ. These findings also support previous studies that show epileptic activity from deep structures can be detected and reconstructed using MEG[37,38]. From the 36 seizures recorded by MEG, 11 seizures do not present identifiable morphological features of ictal activity and hence were not included in the analysis. In all cases there was visibly more muscle artefact contaminating the onset and evolution of the ictal discharge. An example of a failed ViEEG reconstruction, taken from Patient 5 (first seizure), is shown in Supplementary Fig. 1. Ictal noise contamination was highest for Patient 1 (only 1 of 6 seizures reconstructed) and Patient 4 (only 5 of 8 seizures reconstructed). Each patient had at least one seizure that could be reconstructed by ViEEG. Ictal rhythms that were more amenable to ViEEG reconstruction were those with little noise contamination around discharge onset and a clear evolution. In addition to the appearance of the ViEEG waveforms, results only qualified for network analysis if the reconstructed ViEEG signal displayed a paroxysmal disruption of the background waveforms with an evolving ictal-like rhythm. We suspect that compared to source localisation, source reconstruction may require higher SNRs to resolve identifiable ictal features in source space. As well, certain geometries of anatomical structures, such as gyral areas, may impair MEG source reconstruction accuracy. These 11 seizures might be amenable to source reconstruction with corresponding ictal HDEEG signals (the subject of our future work in a larger cohort). Ictal source signals have been reconstructed using ictal

scalp EEG in prior studies[39–41]. However, as opposed to MEG, scalp EEG signals are more distorted when the electrical field propagates through inhomogeneous head tissue. More sophisticated techniques are needed to process and analyse the EEG signals in source space[41].

This study demonstrates proof-of-concept that dynamical network models using ViEEG signals identify a sub-network VIZ that provide a valid characterisation of the EZ and prediction of the clinical localisation (Fig. 3). This finding is significant for the following reasons. First, it demonstrates the feasibility of translating dynamical network models developed from iEEG to non-invasive ViEEG. Second, our proposed approach also identifies non-ictogenic brain areas that are less likely to overlap with the proposed EZ, which may help clinicians fine-tune clinical hypotheses to be tested using invasive approaches. Third, our approach is data-driven and requires less human input compared to routine clinical investigations, making it a more objective assessment. As opposed to the original work of Goodfellow et al.[17] that only analysed the first seizure from each patient, we analysed all seizures that were successfully reconstructed from ictal MEG data.

Ding et al.[39] is the first study to reconstruct ictal EEG source signals to localise the presumed EZ in source space and investigated the causal interaction patterns to identify the primary epileptic sources. Recent work from Lopes et al.[42] and Sohrabpour et al.[41] applied network models and connectivity analysis to EEG source-space networks and demonstrated the feasibility of non-invasive lateralisation and characterisation of the EZ. Motivated by recent studies using invasive iEEG data[17,18,43,44], our work extends the dynamical network models and virtual resection from Goodfellow et al.[17] and Lopes et al.[42] to ictal MEG source signals, which has achieved similar performance in characterising the EZ as Sohrabpour et al.[41] but with a different imaging modality. It is also notable that we analysed the MEG data recorded from complex cases only (MRI-normal or complex lesions) using a ViEEG approach that is intuitive to clinicians

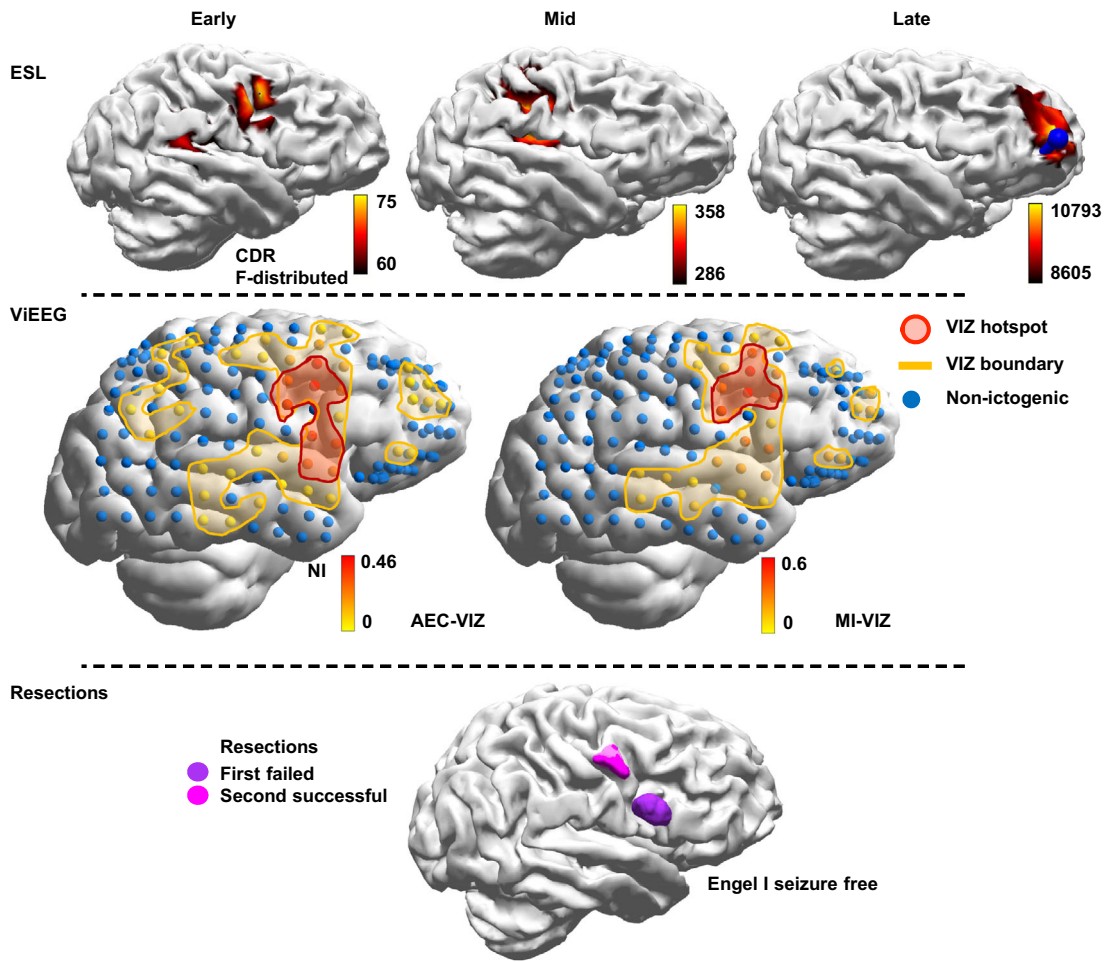

**Fig. 5 (Patient 6) AEC-VIZ and MI-VIZ hotspot concordance with the earliest source localisation solution (early-ESL) and resection margin.** Despite ViEEG being defined by only MSL solutions (only MEG data was used to reconstruct ViEEG signals in this study), dynamical network models suggest AEC-VIZ and MI-VIZ hotspots are concordant with the earliest source localisation solution, which was early-ESL in this case (not early-MSL), and the second resection (magenta). However, note the more dispersed AEC-VIZ NI map that encompasses the first failed resection (purple) as well. Both hotspots are discordant with MSL (see Supplementary Fig. 10 for early-, mid- and late-MSL solutions). Overlap of AEC-VIZ and MI-VIZ boundaries is also present. Note that this patient had focal motor status epilepticus (left face and hand) and a segment of continuous epileptic discharges is modelled. The second resection (years after the first failed resection with normal histology) led to an Engel I seizure-free outcome at follow-up over 2 years (histology cortical dysplasia). *Abbreviations*: MEG magnetoencephalography, iEEG intracranial electroencephalography, ViEEG virtual intracranial electroencephalography, EZ epileptogenic zone, SOZ seizure onset zone, HDEEG high density electroencephalography, VIZ virtual ictogenic zone, MSL MEG source localisation, ESL HDEEG source localisation, AEC amplitude envelope correlation, MI mutual information, AEC-VIZ virtual ictogenic zone using amplitude envelope correlation, MI-VIZ virtual ictogenic zone using mutual information, NI node ictogenicity.

given its resemblance to iEEG arrays. Future work is needed to compare different techniques using EEG and MEG source signals.

The iEEG SOZ is often regarded as a subset of the EZ[45]. This is in part because resection is often performed beyond the extent of the iEEG SOZ and includes non-SOZ electrodes to ensure the removal of the entire putative EZ. However, the resection margin between SOZ and non-SOZ electrodes is often determined by the experience of the treating team which is less objective and less amenable to hypothesis-testing. Recent network studies[17,43] indicate there may exist a regulatory mechanism surrounding the SOZ in the form of pathological dynamics that synchronise and de-synchronise the network and hence regulate seizure generation and propagation. Such regulatory mechanisms merit further consideration in the surgical work-up[43]. Our VIZ, particularly the hotspots, can non-invasively offer such an objective boundary for surgical planning. Further, ViEEG combined with dynamical network models enables hypothesis-testing to assess surgical strategies prior to resection by virtually resecting one or more nodes from the network[17,44,46,47]. For example, all non-seizure-free post-operative patients revealed pre-operative VIZ hotspots that sat outside the resection margins—Patient 1 (Supplementary Fig. 5), Patient 3 (Supplementary Fig. 7), Patient 4 (Supplementary Fig. 8), Patient 9 (Supplementary Fig. 13), Patient 10 (Supplementary Fig. 14), and Patient 11 (Fig. 4, Supplementary Fig. 15).

There has been a long-standing discussion on network models versus source localisation for epilepsy and epilepsy surgery[48,49]. An important question that has been raised is whether network models using EEG and MEG reconstructed source data better characterise the EZ than the solutions offered by source localisation. As demonstrated by Sohrabpour et al.[41], connectivity imaging provides similar accuracy to source localisation in determining the extent of the EZ using sophisticated source imaging algorithms on ictal HDEEG data. Our simultaneously acquired HDEEG–MEG dataset and prospective study validating the clinical utility of electromagnetic source localisation offers the unique opportunity to address such a question[36]. This study compares findings from dynamical network modelling of ictal

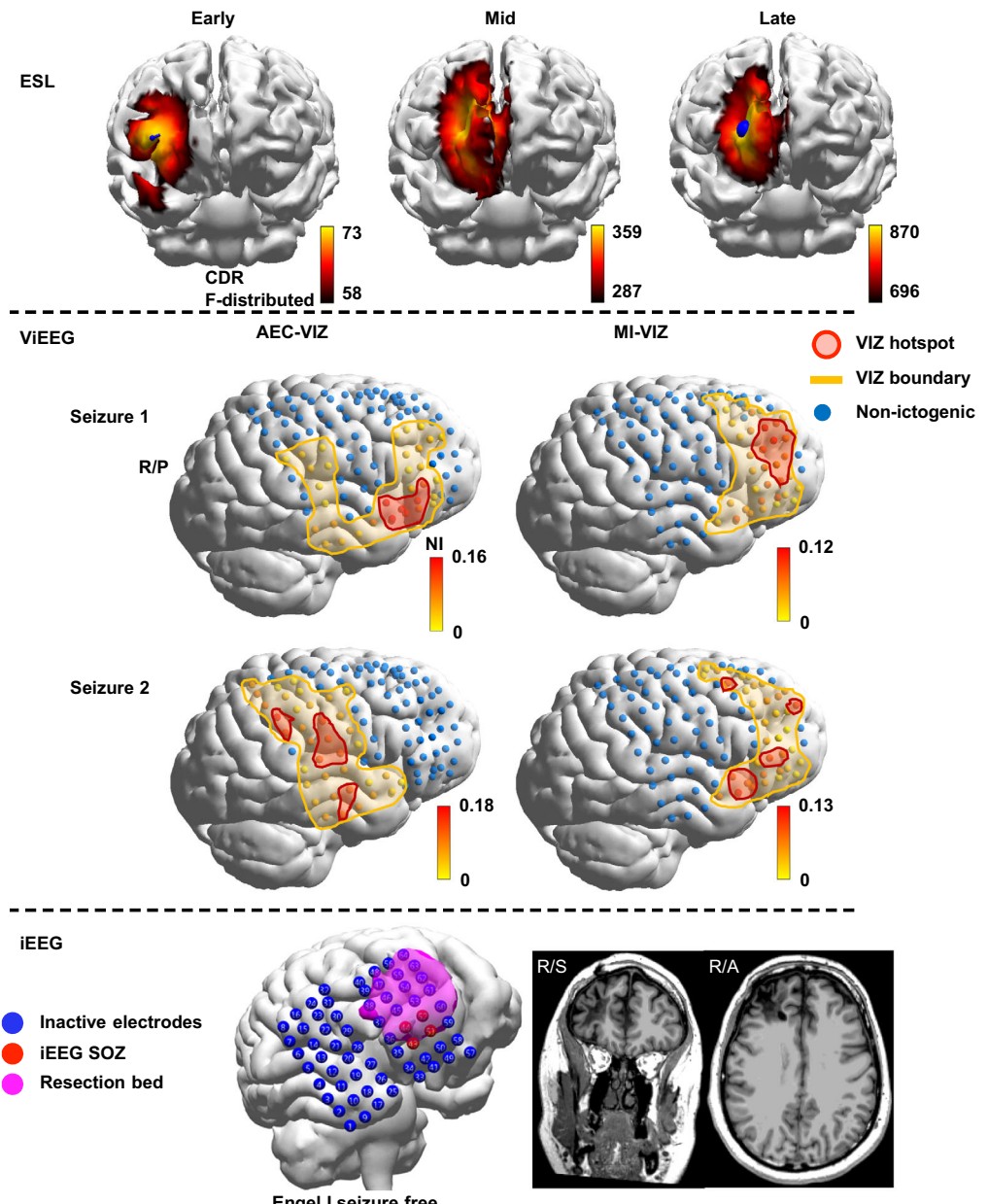

**Fig. 6 (Patient 12) AEC-VIZ is discordant with MI-VIZ and variability is found across seizures.** Dynamical network models identify different hotspots and boundaries between AEC-VIZ and MI-VIZ across two of the recorded seizures. MI-VIZ from seizure 1 gives an extensive boundary in the right frontal lobe with a hotspot that overlaps the resection margin, iEEG SOZ and early-ESL, which is the earliest solution that more accurately predicts the EZ than early-MSL (Supplementary Fig. 16). Boundaries and hotspots for the corresponding AEC-VIZ and MI-VIZ (seizure 2) hotspots are discordant with early-ESL, resection margin and iEEG SOZ. The patient is seizure-free over 24 months post-surgery and hence only MI-VIZ from seizure 1 successfully captures the EZ. *Abbreviations*: MEG magnetoencephalography, iEEG intracranial electroencephalography, ViEEG virtual intracranial electroencephalography, EZ epileptogenic zone, SOZ seizure onset zone, HDEEG high density electroencephalography, VIZ virtual ictogenic zone, MSL MEG source localisation, ESL HDEEG source localisation, AEC amplitude envelope correlation, MI mutual information, AEC-VIZ virtual ictogenic zone using amplitude envelope correlation, MI-VIZ virtual ictogenic zone using mutual information, NI node ictogenicity.

MEG data with source localisation solutions using simultaneous HDEEG and MEG. In our previous work, the earliest sLORETA solution of early-ESL and early-MSL best characterises the EZ than either ESL, MSL, or combined electromagnetic source localisation (EMSL) alone[36]. On combining ictal ViEEG signals from MEG alone with dynamical network models, MI-VIZ predicts the earliest solution that can be only offered by source localisation with two modalities (HDEEG and MEG data). Specifically, MI-VIZ can predict the earliest solution better than predicting the early-MSL solution (Fig. 3A). This finding suggests

dynamical network models can provide valuable information beyond source localisation using a single modality.

We also observe that our proposed approach identifies the wider extent of the VIZ relative to the putative EZ as well as inter-seizure variability. The extent of the VIZ is likely to reflect propagation of seizure activity[50]. Our previous work argues that source localisation of ictal discharges at the mid to late phase of the averaged discharge complex usually reflects areas involved in seizure propagation. Statistical analysis suggests AEC-VIZ and MI-VIZ hotspot and boundary are less likely to predict mid-MSL

and late-MSL (Fig. 3A, Supplementary Tables 2 and 3). Hence, the VIZ is perhaps not a simple representation of seizure propagation networks but its extent could still be affected by seizure propagation. More detailed analysis of seizure propagation networks using iEEG seizures is needed to test this hypothesis.

Two connectivity methods—linear and nonlinear coupling between signals—were employed to better understand the underlying network structures responsible for seizure generation. AEC is believed to solely characterise linear correlations between amplitude envelopes[51,52], while MI measures both linear and nonlinear relationships by quantifying shared and unique information between two time-series[53]. Our results demonstrate that MI-VIZ and AEC-VIZ assist characterising the EZ and predicting the clinical localisation (Fig. 3, Supplementary Fig. 4), which suggests a connectivity approach that captures both linear and non-linear interactions might offer more information about ictal network structures than an approach that only captures one type of interaction. This finding also lends support to a previous theoretical model using nonlinear dynamics from the SOZ to identify the EZ from seizure propagation and predict seizure propagation and termination[16,54].

We find six non-seizure-free patients have at least one VIZ hotspot node outside the resection margin—Patient 1 (Supplementary Fig. 5), Patient 3 (Supplementary Fig. 7), Patient 4 (Supplementary Fig. 8), Patient 9 (Supplementary Fig. 13), Patient 10 (Supplementary Fig. 14), and Patient 11 (Fig. 4, Supplementary Fig. 15)—which has been previously reported using ictal iEEG data[17]. An example is given in Patient 11 (Fig. 4, Supplementary Fig. 15), where both VIZ hotspot and iEEG SOZ are superior to the resection margin and this patient is non-seizure-free with a 60% seizure reduction rate (Engel III). Our results suggest alternative surgical strategies can be devised for non-seizure-free patients with, perhaps, reduced need for invasive monitoring for those patients who are being considered for a second resection. Some VIZs also overlap eloquent cortex as occurs in Patient 10 (Supplementary Fig. 14) and Patient 11 (Supplementary Fig. 15), which is not accounted for in our proposed approach. The balance between seizure outcome and compromise of eloquent cortical function may be integrated into future models to better facilitate decision-making for patients and clinicians.

There are some limitations in this study. Only ictal data were analysed. This is because our dynamical network models investigate the change of cortical excitability that are more likely to be revealed during a seizure state. And while we accept that our seizure rate in the MEG is higher than usual at around 25% of studies performed (12/50 patients), there were several possible explanations. We sleep deprive patients and withhold medication for 12 h before the recording, and we recruit surgical candidates at the more severe end of the seizure frequency spectrum based on the clinical history and the frequency of interictal and ictal events during routine video-EEG telemetry. Nonetheless, given the challenges faced in capturing seizures during a one-hour MEG recording, the next step is to use interictal data from a larger patient cohort to potentially broaden the clinical utility of our method. Our model is phenomenological and does not include details of physiology that underlies the functioning of individual epilepsy patients. Findings from our proposed approach can only serve as a statistical biomarker that offers complementary clinical information to the pre-surgical evaluation. We did not employ connectivity methods that reduce spurious connections (often introduced by volume conduction) in source space to construct functional networks because in our study we looked at dynamical network models to characterise the EZ for epilepsy surgery, instead of investigating neural mechanisms. More details on volume conduction in source-space

networks are discussed in the Supplementary material (Volume Conduction and functional networks). In addition, the computational workload does not allow us to model all the sources (5000–25,000 sources) at once from the whole brain at a spatial resolution (10 mm) comparable to iEEG electrodes. At present, the feasible scale of networks is between 100 and 300 nodes for dynamical modelling and results to be interpreted. In a prospective fashion, ViEEG with a feasible network scale can be moved around freely and modelled iteratively across the whole-brain, although admittedly the ideal scenario is to model a ViEEG covering the whole-brain at once (refer to Considerations for ViEEG locations in Supplementary material). Because we wanted to mimic the ICEEG $10 \times 10$ mm regular array with directly comparable ViEEG $10 \times 10$ mm placements, there were always going to be some ViEEG electrodes that found a position within the mid-point of the gyral crown (where MEG is unlikely to see the ictal discharge). We accept that the choice of a more restricted ViEEG alignment to the outer cortical surface, with incomplete representation of deeper sulcal clefts, reduces the sensitivity of the MEG signals guiding the ViEEG reconstruction. Nonetheless, in spite of the constraint of a subdural grid-like ViEEG configuration that was not ideally suited to detection of tangential sulcal sources, it is encouraging that our ViEEG results were clinically informative. We also recognise that this constraint may also increase a correlation between superficial virtual and subdural electrode findings compared to a situation where ViEEG sources are positioned deeper to the cortical surface. However, because our surgical cohort was primarily investigated with subdural grids at the time, our ViEEG results had to be tested against this reference to satisfy our proof-of-concept study design. We also migrated ViEEG grids to encompass a larger total cortical surface area than covered by the fixed subdural grid locations and, despite this, our VIZ results met statistical significance for predicting the likely EZ. To be clear though, the anatomical extent of ictal generators detectable by MEG is not in the order of millimetres but several centimetres ($3$–$4$ cm$^2$)[55]; thus, with inclusion of multiple peri-sulcal ViEEG positions within the virtual array, we were able to demonstrate the value of this approach for non-invasive estimation of the putative EZ. MEG signal analysis that limits the ViEEG positions to just sulcal and fissural surfaces may well generate more accurate results. We are in a position to do this now with the recent availability at our centre of stereoEEG, which is able to sample sulci more readily as opposed to the grids and strips that were used for the bulk of these cases where ICEEG sampling is limited to ictal activity at the gyral crowns and superficial sulcal crests.

Our study is retrospective with a modest number of patients and seizures analysed compared to studies using iEEG seizures, although it contains one of the highest seizure counts obtained across ictal MEG studies. Our patient cohort was also quite heterogeneous for focal epilepsy sub-type. To our knowledge, to date there are three MEG studies that contain more seizures than our study. Ramanujam et al.[56] described a cohort of 40 patients with at least one seizure but only five patients had undergone surgery (as opposed to all patients in our study). Medvedovky et al.[57] reported 47 patients, but only 11 patients gave a localisable source imaging result (as opposed to our study where all 12 patients gave a localisable result). Alkawadri et al.[58] reported 44 patients who experienced at least one seizure during the MEG recording but, as a retrospective study, surgical follow-up data was only available in 12 patients. The surgical follow-up in our 12 patients was based on prospectively acquired monthly, long-term post-operative seizure counts by Plummer et al.[36]. It is important to note, however, that only one seizure has been analysed at a time, and that multiple seizures (within the same patient), although demonstrated here to give different results, have not

been formally compared. An interesting extension to this work would be to compare different seizures and integrate these results into a patient-specific prediction of the EZ. Our findings motivate further investigation in methodology and clinical utility using multi-centre datasets or prospective studies to devise and optimise surgical strategies objectively and safely.

In conclusion, we have used non-invasively acquired ictal MEG data to reconstruct ViEEG signals with distinctive morphological and spatial characteristics that are comparable to patient-specific invasively acquired ictal iEEG data. VIZ identified by dynamical network models using two connectivity methods (AEC and MI) can predict the resection margin and iEEG SOZ; they also devise an alternative surgical strategy for patients with suboptimal post-operative seizure control. Importantly, VIZ using ictal MEG data alone can predict optimal source localisation solutions that were previously derived from simultaneous HDEEG and MEG data. Our findings suggest ViEEG combined with dynamical network models can provide valuable information beyond source localisation using single modality ictal MEG, thus offering potential clinical utility in the pre-surgical evaluation of complex drug refractory epilepsy cases.

## Methods

**Dataset and ethics**. The study was based on the dataset in Plummer et al.[36]. Patients provided informed consent to participate in the study. Patient consent was obtained according to the Declaration of Helsinki. Ethics approval for the research protocol was given by The Human Research Ethics Committees of St. Vincent's Hospital and Swinburne University of Technology, Melbourne. From 50 consecutively recorded epilepsy surgery candidates, twelve patients had at least one seizure in the MEG scanner (seven males, five females, age range 10–54 years, median 29 years; disease duration 3–32 years). A total of 36 seizures were captured from this cohort who had severe drug-resistant focal epilepsy (daily to weekly seizures and frequent discharges on routine video-EEG telemetry) with either no visible MRI lesion (9 patients) or a complex lesion (3 patients). All patients had at least 20 months post-surgical follow up (median 24 months, range 20–39 months). Each patient had an hour-long simultaneous HDEEG (ANT Waveguard® 72–94 electrode cap including 12-electrode inferior temporal array, 10–10 positions) acquired on one of two EEG amplifier systems (ANT ASAlab®, Enschede; Compumedics SynampRT®) and MEG (Elekta Triux® 306 sensors, 102 magnetometers, 204 planar gradiometers), sampled at 1000 or 5000 Hz with anti-aliasing filter set at 330 or 1650 Hz, respectively. Bad channels for all MEG data were checked prior to applying temporal extension to signal source separation (tSSS) using Maxfilter® v2.2.10–15 (Elekta Oy, Helsinki) for interference suppression (correlation limit 0.98 and sliding window of 10 s). Independent one-second interval clock triggers acquired on each system were used to synchronise HDEEG and MEG offline, verified by comparing ECG channel signal phase from each independent modality. MEG head coils, HDEEG electrode positions, and PAN (pre-auricular, nasion) coordinates were digitised (Polhemus Fastrak®) in common space for MRI coregistration; digitised points were cross-validated with optical sensor tracking (NDI Polaris Vicra®). Applying a second-order bandpass Hann FFT filter (1–100 Hz), seizures were independently marked by a neurologist (C.P.) and a clinical scientist (S.V.) using Curry 7® (Compumedics Neuroscan®, Hamburg). A total of 36 ictal MEG events were captured from 12 patients (interictal-only activity in one MRI-normal patient) (Table 1). Pre-operative MPRage MRI was acquired as part of clinical investigation as well as post-operative CT to confirm resection margins. A post-iEEG implantation CT scan confirmed implantation locations in the seven patients who had iEEG grids or strips— Patient 7 had an intraoperative depth electrode and four patients (Patients 2, 3, 6, 8) went straight to surgery based on non-invasive data. As raw iEEG recordings were not available to us for two external patients (only iEEG reports for Patients 7, 9) we were able to compare ViEEG to iEEG results in six patients (Patients 1, 4, 5, 10, 11, 12). Based on findings from Plummer et al.[36], ictal early-MSL preceded early-ESL in six patients (Patients 1, 3, 4, 5, 10, 11), while ictal early-ESL preceded early-MSL in five patients (Patients 2, 6, 8, 9, 12) Ictal discharges were not localisable in one patient with either ESL or MSL (Patient 7) (Table 1). Using standardised low resolution tomographic analysis (sLORETA), Plummer et al.[36] found there is usually a lead–lag relationship between early-MSL and early-ESL for a given averaged ictal discharge and it was the earliest solution (defined as the first occurring early-MSL or early-ESL solution from discharge take-off that explained 90% of the signal variance) that was the best predictor of the proposed EZ.

**Virtual iEEG**. We propose a concept (ViEEG) that consists of multiple virtual electrodes or virtual sensors guided by MEG source imaging[35,59]. While our original source imaging study[36] used HDEEG and MEG data, the present study only uses MEG data. This is because of the added complexity of combining

HDEEG–MEG signals in the same source space given the relative differences between modalities for ictal onset latencies and tissue conductivity effects. Here, the simultaneously acquired HDEEG was used to help confirm an ictal MEG rhythm. An example of the MEG-informed ViEEG workflow for a patient is given in Fig. 1. ViEEG was defined using similar electrode configurations to iEEG, such as grid, strip and depth arrays, and a uniform spacing (10 mm) between virtual electrodes to allow a direct comparison between iEEG and non-invasive ViEEG. All cortical reconstructions are rendered 50% transparent to permit visualisation of the ViEEG electrodes so that they are actually deeper than they appear (they sit within the bed of the cortical ribbon and not superficially at the dural surface). Given the higher sensitivity of MEG to sulci and fissures, ViEEG electrode arrays were positioned to optimise coverage of the deeper peri-sulcal cortical ribbon and fissural surfaces within the constraints of this array with 10 mm × 10 mm inter-electrode distances. We configured the ViEEG array in this manner to mimic the array given by the subdural grid to allow a more direct correlation of ictal waveforms between actual and virtual intracranial signals across comparable sensor geometries. ViEEG positions, however, were not based on exact iEEG locations (operator blinded to these locations) but were instead freely mobilised around the cortex to encompass and to extend beyond all MSL solutions ViEEG was, therefore, defined for each patient using information from MSL[36] and not ESL in order to limit any subjectivity tied to the manual selection of ViEEG locations. Thus, the locations of ViEEG electrodes extensively covered MSL of averaged ictal discharges (early-, mid-upstroke and late-peak phase solutions)[36]. Both the phase order of the MSL solutions and the solution modality (ESL or MSL) that gave the best predictor of the proposed EZ in a given patient were not known when ViEEG locations were defined. Further, the ViEEG was set up to include the resection volume well within its boundaries with prior knowledge of the lobar region of interest containing the resected volume, but without knowledge of the specific configuration of the resection bed. ViEEG comprising over 500 virtual electrodes is less computationally feasible for dynamical modelling and so we did not define ViEEG to cover the whole brain in this study. We used Curry 8® (Compumedics Neuroscan®, Hamburg) software and pre-operative MRI scans to generate realistic boundary element method models with a single layer, i.e., inner skull surface[60]. Ictal ViEEG signals were reconstructed using a linearly constrained minimum variance beamformer technique[61]. Technical details of MEG signal processing and source reconstruction are presented in Supplementary material (ViEEG signal reconstruction). A total of 25 from 36 seizures could be reconstructed using ViEEG with distinct morphological and spatial characteristics from background activity (Supplementary Table 6, Supplementary Fig. 17). Eleven seizures reconstructed from the ictal MEG data did not present identifiable morphological features of epileptiform discharges and hence were excluded from analysis (Supplementary Fig. 1). ViEEG configurations are presented in Supplementary Figs. 5–16.

**Blinded analysis**. To ensure the validity of ViEEG and dynamical network models, M.C. was blinded to the surgical outcomes, pathology, iEEG findings, and ESL solutions; M.C. had no knowledge of which solution (early-ESL or early-MSL) was the earliest solution for each patient when defining ViEEG. All other team members were blinded to the locations and configurations of ViEEG. When analysing and modelling ictal ViEEG signals, D.G. was blinded to clinical information, including resection margins, iEEG findings, source localisation and surgical outcomes, and ViEEG locations and configurations. Only time-series of ictal ViEEG signals (no information on ViEEG locations, resection margin, pathology or surgical outcomes) were given to construct functional networks and apply dynamical network models. Patient information was de-identified and patient numbers were randomised from the previous publication[36]. Clinical information and ViEEG configurations were only unblinded to D.G., W.W. and J.R.T. after results from network models were finalised.

**Dynamical network models and virtual resection**. Functional networks were constructed from ictal ViEEG signals using two connectivity methods, amplitude envelope correlation (AEC) and mutual information (MI)[17,23,62–64]. Both methods do not conservatively correct field leakage effects in reconstructed source signals[63]. Similar to connectivity analysis of iEEG data, we treated each ViEEG electrode as a node and connectivity between a pair of nodes expressed by the edge-strength. A surrogate correction threshold was then applied to functional networks to preserve edges that were statistically significant[64]. Next, a generative dynamical network model was integrated to each node, simulating the change of network excitability when a node was virtually resected from the network[17,65]. Specifically, we calculated the change in time that the network spends in the seizure state over the total simulation time after a node is removed from the network structure. The effect of the removal of a node is quantified by the so-called node ictogenicity (NI), where a positive NI indicates the removal of that node effected a decrease in the seizure-likelihood. This model has been presented previously[17,42,46,66]. The set of nodes that have significantly large (Mann–Whitney–Wilcoxon $U$-test and Bonferroni–Holms correction survival) and positive NI values relative to other nodes is defined as the virtual ictogenic zone (VIZ). Any nodes outside the VIZ are defined as non-ictogenic nodes. For each MEG seizure, our approach identified two VIZs (AEC-VIZ and MI-VIZ) from AEC and MI constructed networks. Each VIZ also has two aspects, hotspot and boundary. We defined a VIZ hotspot as nodes in the top 20% NI values among all VIZ nodes and boundary as the outer line of the

VIZ. The VIZ hotspot (or top NI value nodes) has been shown to better predict the likely EZ, while the VIZ boundary can capture the entirety of the EZ and identify non-ictogenic brain areas that are less likely to overlap with the EZ[17,42,46]. See also Supplementary material (Network model).

**Statistical analysis**. A mixed-effects logistic regression model was employed to evaluate the statistical significance of nodal-level (virtual electrode) concordance between a VIZ (hotspot and boundary) and clinical localisation (resection margin, iEEG SOZ and HDEEG–MEG source localisation) based on odds ratios (OR) with 95% confidence intervals (CI). Akaike information criterion (AIC)[67] and Bayesian information criterion (BIC)[68] were reported. We also calculated the precision and recall of respective boundaries and hotspots of AEC-VIZ and MI-VIZ in predicting the resection margin and the earliest source localisation solution when seizures are grouped based on long-term (median 24-month follow-up) surgical outcomes (seizure-free and non-seizure-free). We applied the combination of Engel score (as a largely qualitative measure) and the quantitative measure of percentage seizure change (as determined prospectively in our original study)[36] to classify patients as genuinely seizure-free or not. We propose that this combined outcome measure is more discriminative when linking the resection bed to the notional EZ in seizure-free patients. F-scores, the harmonic mean of precision of VIZ hotspot and recall of VIZ boundary, were reported. See also Supplementary material (Statistical analysis).

**Reporting summary**. Further information on research design is available in the Nature Research Reporting Summary linked to this article.

## Data availability

All data are available upon request. Source data are provided with this paper. Simulation dataset (from Supplementary File) is available in the repository under 'supplemental_materials_ground_truth'. Specifically, the raw MEG data are available upon request for reasons of patient confidentially. Processed clinical data are provided in the Supplementary Information/Source Data file.

## Code availability

Codes used in this study have been deposited at the following publicly available address https://cloudstor.aarnet.edu.au/plus/s/Ifl2zmMAWvuj4A5.

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

## Acknowledgements

The authors acknowledge the facilities, and the scientific and technical assistance of the National Imaging Facility at the Swinburne Node, Swinburne University of Technology with particular thanks to Dr. Rachel Batty, Ms. Mahla Cameron-Bradley, and Ms. Johanna Stephens for their technical support. We acknowledge the Australian National Imaging Facility for the support of W. Woods and the MEG system at Swinburne University of Technology. We acknowledge the neurologists Dr. Simon Harvey (Royal Children's Hospital Melbourne), A/Prof. John Archer (Austin Hospital Melbourne), A/Prof. Wendyl D'Souza (St. Vincent's Hospital Melbourne), A/Prof. Ross Carne (St. Vincent's Hospital Melbourne), and the neurosurgeons A/Prof. Michael Murphy (St. Vincent's Hospital Melbourne), Mr. Kristian Bulluss (St. Vincent's Hospital Melbourne), and Ms. Wirginia Maixner (Royal Children's Hospital Melbourne) whose patients were included in the study. The authors would also like to thank Prof. Thomas Knosche, Dr. Christian Rummel and Dr. Marinho Lopes for their insightful discussions on MEG source imaging and dynamical models. M.C. acknowledges the support from Australian Government Research Training Scholarship and St. Vincent's Health Foundation, Australia. M.J.C., A.P. and J.R.T. acknowledge the financial support from the Royal Society International Exchanges Award (grant number IE170112). D.G. and J.R.T. acknowledge the generous support of the Wellcome Trust Institutional Strategic Support Award (grant no. 204909/Z/16/Z). W.W. acknowledges the financial support of the MRC via grant (MR/N01524X/1) and Epilepsy Research UK via Grant (F2002). J.R.T. acknowledges the financial support of the EPSRC via grants (EP/N014391/2 and EP/T027703/1). F.W. acknowledges the support from Youth Innovation Promotion Association at the Chinese Academy of Sciences via grant (2019096).

## Author contributions

Study design: M.C., S.V., W.P.W., A.P., C.P., M.J.C.; Data acquisition: M.C., S.V., C.P. W.P.W.; Analysis: M.C., D.G., S.J.V., W.P.W., S.V., F.W., W.W., J.R.T., A.P., C.P., M.J.C.; Writing original draft: M.C., C.P.; Writing review and editing: M.C., D.G., S.J.V., W.P.W., S.V., F.W., W.W., J.R.T., A.P., C.P., M.J.C.; Supervision: J.R.T., A.P., C.P., M.J.C.

## Competing interests

The authors declare no competing interests.
