## [Peer Review File · Nature Communications]

Virtual intracranial EEG signals reconstructed from MEG for epilepsy surgeryREVIEWER COMMENTS

Reviewer #1 (Remarks to the Author):

Reviewing "Virtual intracranial electroencephalography for epilepsy surgery using ictal magnetoencephalography"

In this paper, the authors propose a novel framework to identify the epileptogenic zone (EZ) in focal epilepsy patients. This framework employs non-invasive electromagnetic scalp measurements, in this study MEG, to identify the EZ using source imaging algorithms to estimate underlying sources' time-course of activity at selected location, i.e., virtual electrodes, then the connectivity among different locations is calculated and a neural mass model, the theta model, is run to ultimately determine the epileptogenic nodes at the virtual electrode locations. In this manner, the EZ can be objectively determined, noninvasively. The authors have tested the proposed algorithm in 13 patients and demonstrate good concordance between estimated EZ and clinical findings. The theory is well thought and developed, the study well performed and the manuscript well written. However, I have a few concerns and suggestions that I think can help improve this interesting work which I enjoyed reading.

Major Issues.

1. Selection of Virtual Electrode Locations – Consideration of Surgical Margins. It is mentioned in the paper that the locations for the virtual electrodes were selected based on the ESL and MSL solutions and the researchers were blinded to clinical results and SOZ electrodes, but resection margins were considered. This seems odd to me. How is that possible if the researchers were blind? Please fully and clearly explain this process. Additionally, would it not be better to solely choose the electrode locations based on the source imaging algorithm to ensure the objectivity of the whole method which is based on non-invasive measurements and information prior to intervention/surgery? IN this manner it is more likely that the method can be clinically used to predict the EZ or help shape the medical intervention.
2. Patient and Seizure Number. The number of patients and seizures studied in this study is limited, and if possible it must be increased for a more rigorous validation, but I understand that recording ictal activity in MEG is not easy.
3. Considerable Number of Removed Seizures - I. About 11 out of 36 seizures were not included in the study as the source imaging results did not work well and the estimated time-courses did not seem realistic. How did you determine this low quality? I am aware of the conditions you have proposed (based on visual inspection), in the supplementary, but isn't that subjective? Can you present an example of the "failed" seizures so that we can also see what went wrong?
4. Considerable Number of Removed Seizures - II. Following my previous comment, how did you determine the good quality of the seizures you kept? I believe you have the intra-cranial EEG recordings in 9 of these patients; given that typical seizures have more or less the same properties (given that iEEG and E/MEG might not have been recorded simultaneously), can you use objective measures such as correlation, spectral features, etc. to assess how well the estimated sources match/mismatch the underlying sources? Is there a more objective method to make this determination?
5. Considerable Number of Removed Seizures - III. It seems that most of these discarded seizures (8 out of 11) are coming from patients 1 and 4. Are these all typical seizures in these patients? I mean from the 6 seizures recorded in patient 1, are they all the typical seizures observed in this patient? Or the 5 that "failed" are another type? Basically, have you looked more carefully into the reason these seizures "fail"? Is the SNR or recording quality the problem? Do they belong to another category compared to the seizures that worked?
6. Epileptogenic Zone. The authors use the term EZ interchangeably with resection (if my understanding is correct). If all the patients studied here were seizure-free we could take the resection as the pseudo-EZ as the removal of those regions has stopped seizures, but the study includes non-seizure-free patients (which might be due to not fully treating potential EZ suggested

by your algorithm). I think you might want to either caution about this in the manuscript or use another term such as "proposed EZ" or "clinically determined/proposed EZ".

Suggestion - 1. One of the shortcomings of beamforming methods, specifically older versions, is that correlated and coherent sources are not separated/estimated well, which is a potential issue in ictal imaging as you have discussed in the manuscript and the supplementary. However, I believe you can assess this with some simulations in your work. You can randomly assign some nodes of activity in the brain, assign some of them as the EZ, run the theta model with some connectivity matrix, run the forward problem to simulate MEG measurements at some frequency bands (add interference and noise at source level and sensor level) and perform your proposed algorithm on this simulated data to assess how different source configuration, interference/noise level and its distance to EZ, effect of virtual electrode selection (during the inverse process), etc. affect your results specifically the beamforming portion of it. The whole framework in its entirety has not been checked yet (even though some of the parts have been separately tested in your previous publications).

Suggestion - 2. Is it possible to share your codes in this new framework, from A to Z, specifically with simulation data (that does not have the sensitivity of clinical data) to benefit the community?

Minor Issues.

1. ViEEG Numbers. Could you kindly include the number of ViEEG electrodes for each patient as well as the number of iEEG electrodes in some existing/new table? I know that you do not increase 500 based on the manuscript, but I was not sure if you fixed your electrode numbers to 500 or some number or not. I was thinking that if it is based on E/MSL size (plus resection margin), it will vary and since you are keeping the distance as 1 cm, the number will be different for each patient.

2. Brain View. The brain sub-figures containing clinical findings such as resection and SOZ, are different from the E/MSL figures, particularly the view. It is not easy to visually check the resection on top of your results. Is it possible to keep the view (brain orientation, view, etc.), the same for all sub-figures of all figure, so that it is easier to visually inspect the results? For instance, look at Fig. 3.

3. E/MSL Threshold of 0.8. How was the threshold of 0.8 chosen for sLORETA results? Regarding point 1, can you change this threshold for ViEEG location selection so that you do not rely on surgical resection margin, at all?

4. Typo in Supplementary. In page 2 of supplementary line 50 I believe it should read "... because removing it has the effect ..." which currently is "...removing it as the effect ...".

Reviewer #2 (Remarks to the Author):

This paper presents an original and interesting approach to finding the epileptogenic zone from the analysis of seizure data recorded with MEG. Using a network analysis and a virtual resection approach, the method provides a definition of the EZ. This definition is compared to the more traditionally defined SOZ (from MEG or EEG source modeling) and to the resection in 12 operated patients. Results indicate a solid predictive value for the EZ defined by the network approach. The paper is not easy to read in part because the method is complex. The paper cannot be read without frequent reference to the supplementary material, which is very extensive.

A significant weakness of the paper is the small number of patients. I realize that patients with seizures recorded during MEG are infrequent but this very fact reduces considerably the applicability of the method (see below). There are only 6 seizure-free patients. Many researchers would have considered all patients with Engel class I outcome in one group, and I assume that given the size of this group, results would have been quite different then.

I was surprised to see that all so-called virtual EEG locations were located on the outer mantle of the brain, in the same region as subdural electrodes are located. Source analysis is usually performed on the surface of the cortex, including in sulci. This is particularly striking because MEG is especially sensitive to generators located in sulci and less so to generators located in gyri. The

authors should justify this choice and should probably change their nomenclature: the virtual electrodes are in the location of subdural electrodes, mostly on the inner surface of the skull; the term "intracranial" is more general and I certainly had expected virtual electrodes within sulci of in the depth of the brain when reading the title and the abstract.

The patient selection process must be described in detail to be able to judge the generalizability of the results. It is unusual to record seizures during a 1-hour MEG recordings and 12 of the 13 studied patients had seizures, in fact quite a few seizures. Among how many consecutive MEG recordings were these patients selected and on what basis? In my experience, it is one every 15 or 20 patients who has a seizure during a MEG study. The general pool of patients from which these were selected must be described. The discussion must include a clear statement regarding how often this method is applicable in a group of patients who are candidate for epilepsy surgery. Two apparently contradictory statements are given: "ViEEG was defined for each patient solely using information from MSL31 (not ESL)" and "M.C. had no knowledge of which solution (early-ESL or early-MSL) was the earliest solution for each patient when defining ViEEG". This needs an explanation.

I am confused by the statement on lines 228 and following: the authors write that the real iEEG and the virtual iEEG signals have a similar distribution. I find them very different in their morphology and spatial distribution. I am not sure how to compare locations though. Unfortunately the supplementary material does not show real iEEG seizures to allow comparison in other cases. Making such a statement and providing only this one non-convincing picture is not appropriate. The authors talk about virtual iEEG and there should be information on how the virtual EEG resembles the actual iEEG. Given the apparent lack of resemblance and the absence of demonstration of this resemblance, it is more appropriate to talk about a method to predict the EZ, but not about virtual EEG.

Figure 3. Such a case does not support a more extended SOZ. It only states that the resected region is not the epileptogenic zone, which could be small but located in a different region. This is not known.

In the figures, the resection should be shown on the same map as the hotspot of the VIZ; otherwise they are difficult to compare. The boundaries of the VIZ do not seem very meaningful and could be ignored if the figure becomes too complex.

The F-score is used to report the results in the main part of the manuscript but this is not easy to interpret, particularly as it incorporates the VIZ boundary, which does not seem to be very meaningful. I think it would be more informative to have the precision and recall results, which are more practically interpretable (now in supplementary material).

The authors conclude that the non-linearly defined networks give better results than the linearly defined networks, and make other such comparisons. Given the small number of patients and the large scatter of the results, I do not think it is appropriate to make such conclusions when there are no statistical comparisons between the measures.

The authors write "This lends confidence to our approach, as consistency of VIZ hotspots was found between seizures for a given patient – refer to Patient 2 (Supplementary Fig. 6), Patient 8 (Supplementary Fig. 12), and Patient 11 (Supplementary Fig. 15)". There is no quantification of the overlap from seizure to seizure and there are patients for whom there is very poor overlap. The authors should either remove this statement or also include the cases of discordance and discuss them. On the subject of multiple seizures, this may be explained somewhere, but it is not clear to me how the study dealt with the several seizures of one patient when making the patient-based predictions.

Reviewer #3 (Remarks to the Author):

The authors propose a combination of MEG source localization and dynamic network modeling to create virtual intracranial EEG (ViEEG) solutions at the cortical level that can estimate virtual ictogenic zones (VIZs) from non-invasive ictal MEG recordings. They consider two measures of connectivity to be used for simulating network activity using the theta model, namely Pearson correlations of the envelopes and mutual information. They comprehensively validate the estimated VIZ hotspots and boundaries, and demonstrate high precision/recall with respect to the clinically detected epileptogenic zones.

This is quite a well-written manuscript, the results are impressive, and the methods are thoroughly thought out, applied, and carefully explained. In particular, the results serve as a stepping stone for several future studies enabling further clinical validations and using both MEG and EEG to improve ViEEG estimation. I have the following comments for the authors:

1. While references 23 and 24 (lines 79 and 81) are relevant to the description of EEG and MEG and their shortcomings, I suggest that the authors add a few more standard references and/or surveys for EEG and MEG source localization as well. In particular, 23 is more relevant to magnetic shielding (than the nature of EEG and MEG signals and their relation to cortical activity). Some suggestions: Hamalainen et al. (1993), Baillet et al. (2001), Schoffelen & Gross (2009), Hari & Puce (2017), etc.

2. The authors give a compelling argument in the discussion that using MEG (instead of EEG) is preferable due to less sensitivity to the type of conductance-based head model used, but all along I was wondering why the LCMV was not jointly applied to MEG + EEG data. I suggest adding a justification to this effect in the methods section Virtual iEEG (lines 153-154).

3. While the explanation of the Blinded Analysis is worthwhile, as it suggests that possible subjective biases have been eliminated from the analysis, it may be counterproductive to more complex follow-up studies. The full knowledge of the data and clinical attributes by all the team members can help optimize the choice of the parameters, models, etc. The subjective biases in this case can be avoided by the common separation of the data into training and testing sets. I suggest that the authors give a more clear justification of why such blinded analysis was necessary.

4. In Fig. 2, the iEEG traces are much more spatiotemporally localized than the ViEEG traces, i.e., the temporal patterns of the hippocampal sources are quite distinct from the basal and lateral sources. This is not the case for ViEEG traces, for which the temporal patterns seem more correlated across the cortical areas. Is this a result of the source mixing due to poor localization? Please clarify.

5. The examples referred to on lines 277-278 are a bit unclear. It seems that Patient 6 is an example of AEC-VIZ being a better predictor of iEEG SOZ, and Patient 12 seizure 1 is an example of AEC-VIZ being a better predictor of mid-MSL. But it is stated that both are examples of MI-VIZ hotspots being better predictors of the resection margin and the earliest solution. While the latter is the case for Patient 12, it doesn't seem to hold for Patient 6. Please clarify. Also, the supplementary figures are mis-numbered: please change Supp. Fig. 10 to 9, and Supp. Fig. 16 to 15.

6. Supplementary material, line 11: please add a reference for the amplitude adjusted Fourier transform surrogate generation.

7. Supplementary material, line 28: The first term on the right hand side of the differential equation for θ_j must be $(1-\cos(\theta_j))$. The symbol θ_j is dropped. Please revise.

8. Supplementary material, lines 37-41: How is the "seizure state" defined for simulated activity using the θ model? Is it based on thresholding the amplitude of each node? Please explain.

9. Supplementary material, lines 184-200: the AIC and BIC are reported, but it is not clear which criterion was used to determine the model order in the logistic regression. Please clarify.

10. Line 424: Please change "ViEEG with on feasible network scale" to "ViEEG with a feasible network scale"

RESPONSE TO REVIEWER COMMENTS

Reviewer #1 (Remarks to the Author):

Reviewing “Virtual intracranial electroencephalography for epilepsy surgery using ictal magnetoencephalography”

In this paper, the authors propose a novel framework to identify the epileptogenic zone (EZ) in focal epilepsy patients. This framework employs non-invasive electromagnetic scalp measurements, in this study MEG, to identify the EZ using source imaging algorithms to estimate underlying sources’ time-course of activity at selected location, i.e., virtual electrodes, then the connectivity among different locations is calculated and a neural mass model, the theta model, is run to ultimately determine the epileptogenic nodes at the virtual electrode locations. In this manner, the EZ can be objectively determined, noninvasively. The authors have tested the proposed algorithm in 13 patients and demonstrate good concordance between estimated EZ and clinical findings. The theory is well thought and developed, the study well performed and the manuscript well written. However, I have a few concerns and suggestions that I think can help improve this interesting work which I enjoyed reading.

Response: Thank you for the comments.

Major Issues.

1. Selection of Virtual Electrode Locations – Consideration of Surgical Margins. It is mentioned in the paper that the locations for the virtual electrodes were selected based on the ESL and MSL solutions and the researchers were blinded to clinical results and SOZ electrodes but resection margins were considered. This seems odd to me. How is that possible if the researchers were blind? Please fully and clearly explain this process.

Response: In Supplementary material (Lines 88 to 92), we state that “ViEEG locations are defined to extensively cover the early-, mid-, and late-phase MSL solutions as well as the entirety of the resection margin. It is important to note that the choice of location, shape and orientation of ViEEG does not take into account any other information (such as shape of resection or pathology).”

To further clarify, ViEEG locations were set by M.C. blinded to iEEG array location, SOZ electrode locations, post-operative outcome, configuration of the resection margins, and

pathology). Because we wanted to include the resection volume well within the frame of the ViEEG set-up, M.C. was aware of the lobar region of interest containing the resected volume, but not the specific sublobar configuration of the resection bed. We have added expanded on this point in the Main paper now (Lines 169 to 171).

“Further, the ViEEG was set up to include the resection volume well within its boundaries with prior knowledge of the lobar region of interest containing the resected volume, but without knowledge of the specific configuration of the resection bed.”

While M.C. was aware of the locations of the MEG Source Localisation (MSL) given by sLORETA, it is important to note that each MSL solution had 3 sub-solutions based on the time-point used for sLORETA modelling of the averaged ictal discharge – early, mid, and late (Plummer et al, 2019). ‘Early’ was the earliest latency sLORETA MSL solution that reached 90% of the measured signal variance from discharge take-off; ‘Mid’ was the MSL solution at the half-way time-point between take-off and the averaged ictal discharge peak; ‘Late’ was the MSL solution at the time-point of the peak. Plummer et al. (2019) found that the early solution was a better predictor of the epileptogenic zone (EZ) than the mid and late solutions and that the locations of the 3 solutions were typically quite dispersed (arguably from cortico-cortical propagation). While author M.C. placed the ViEEG to cover all 3 solution locations, M.C. was blinded to the phase information (that is, M.C. did not know which of the 3 solutions was the early solution). Please also note that M.C. was blinded to and did not use the corresponding ESL solutions to guide the placement of the ViEEG. Indeed, Plummer et al. (2019) found that the early-ESL solutions were a better predictor of the EZ than corresponding early-MSL solutions for ictal discharges.

We have amended Line 163-168: “ViEEG was, **therefore**, defined for each patient using information from MSL³⁶ and not ESL **in order to limit any subjectivity tied to the manual selection of ViEEG locations**. Thus, the locations of ViEEG electrodes extensively covered MSL of averaged ictal discharges (early-, mid-upstroke and late-peak phase solutions. **Both the phase order of the MSL solutions and the solution modality (ESL or MSL) that gave the best predictor of the proposed EZ in a given patient were not known when ViEEG locations were defined**”

We have also amended line 153 to avoid confusion as this is a MEG only analysis: “We propose a novel concept (ViEEG) that consists of multiple virtual electrodes or virtual sensors guided by ~~HDEEG~~ and MEG source imaging “

We reiterated the blinding process in Lines 185-195: “To ensure the validity of ViEEG and dynamical network models, M.C. was blinded to the surgical outcomes, pathology, iEEG findings, and ESL solutions; M.C. had no knowledge of which solution (early-ESL or early-MSL) was the earliest solution for each patient when defining ViEEG. All other team members were blinded to the locations and configurations of ViEEG. When analysing and modelling ictal ViEEG signals, D.G. was blinded to clinical information, including resection margins, iEEG findings, source localisation and surgical outcomes, and ViEEG locations and configurations. Only time-series of ictal ViEEG signals (no information on ViEEG locations, resection margin, pathology or surgical outcomes) were given to construct functional networks and apply dynamical network models. Patient information was de-identified and patient numbers were randomised from the previous publication. Clinical information and ViEEG configurations were only unblinded to D.G., W.W. and J.T. after results from network models were finalised.”

Additionally, would it not be better to solely choose the electrode locations based on the source imaging algorithm to ensure the objectivity of the whole method which is based on non-invasive measurements and information prior to intervention/surgery? IN this manner it is more likely that the method can be clinically used to predict the EZ or help shape the medical intervention.

Response: Yes, we agree that an entirely non-invasive guidance of the ViEEG positions with MSL and not iEEG locations is ideal. And this is what we did. While the ViEEG ‘grids/strips/depths’ look like they have been configured to mirror the locations of the iEEG, this is not really the case. We apologise for contributing to any confusion here. We have clarified this by amending Lines 157 to 163: “An example of MEG-defined ViEEG for a patient is given in Fig. 1. ViEEG was defined using similar electrode configurations to iEEG, such as grid, strip and depth arrays, and a uniform spacing (10mm) between virtual electrodes to allow a direct comparison between iEEG and non-invasive ViEEG. **ViEEG positions, however, were not based on iEEG locations (operator blinded to these locations) but were instead freely mobilised around the cortex to broadly encompass all MSL solutions.**”

2. Patient and Seizure Number. The number of patients and seizures studied in this study is limited, and if possible it must be increased for a more rigorous validation, but I understand that recording ictal activity in MEG is not easy.

Response: Yes, we agree that the study has a limited number of patients. The COVID pandemic has meant that our MEG facility has been in lockdown for over 18 months and hospital waiting times for standard inpatient epilepsy surgery work-up and elective operative lists have ballooned. Because we also need to follow up these patients for at least 12 months after their surgery to assess their outcome, the recruitment of new patients would potentially extend this work by another two years. We would stress that this was a proof of concept study as a possible prelude to a larger clinical study. Our patient cohort was quite heterogeneous for focal epilepsy sub-type and, to our knowledge, there are only a few MEG studies that contain more seizures than our study. We have included these points in the section on Limitations (Lines 476-484). Our patient cohort was also quite heterogeneous for focal epilepsy sub-type. To our knowledge, there are three MEG studies that contain more seizures than our study. Ramanujam et al.⁶⁵ described a cohort of 40 patients with at least one seizure but only five patients had undergone surgery (as opposed to all patients in our study). Medvedovsky et al.⁶⁶ reported 47 patients, but only 11 patients gave a localizable source imaging result (as opposed to our study where all 12 patients gave a localizable result). Alkawadri et al.⁶⁷ reported 44 patients who experienced at least one seizure during the MEG recording but, as a retrospective study, surgical follow-up data was only available in 12 patients. The surgical follow-up in our 12 patients was based on prospectively acquired monthly, long-term post-operative seizure counts by Plummer et al.³⁶.

3. Considerable Number of Removed Seizures - I. About 11 out of 36 seizures were not included in the study as the source imaging results did not work well and the estimated time-courses did not seem realistic. How did you determine this low quality? I am aware of the conditions you have proposed (based on visual inspection), in the supplementary, but isn't that subjective? Can you present an example of the "failed" seizures so that we can also see what went wrong?

Response: As discussed in the main paper, not all MEG seizures were reconstructed with clear ictal waveforms or a distinct transition to seizure state. We believe the main reason for the failed ViEEG reconstruction was the relatively low SNR characteristics of those 11 events. A clear pattern emerged whereby early onset of sustained muscle artefact and briefer ictal runs appeared to limit the reproducibility of source reconstruction in these cases. An example of a failed seizure is given in Supplementary Fig 1, where the second MEG seizure from Patient 5 was reconstructed using ViEEG but did not present clear ictal waveforms or a

distinct transition to the seizure state. Ictal ViEEG signals reconstructed from the first MEG seizure for Patient 5 is shown in Supplementary Fig. 9.

We have also amended the paragraph (Line 332 to 342): “From the 36 seizures recorded by MEG, 11 seizures do not present identifiable morphological features of ictal activity and hence were not included in the analysis. **In all cases there was visibly more muscle artefact contaminating the onset and evolution of the ictal discharge. An example of a failed ViEEG reconstruction, taken from Patient 5 (first seizure), is shown in Supplementary Fig. 1. Ictal noise contamination was highest for Patient 1 (only 1 of 6 seizures reconstructed) and Patient 4 (only 5 of 8 seizures reconstructed). Each patient had at least one seizure that could be reconstructed by ViEEG. Ictal rhythms that were more amenable to ViEEG reconstruction were those with little noise contamination around discharge onset and a clear evolution. In addition to the appearance of the ViEEG waveforms, results only qualified for network analysis if the reconstructed ViEEG signal displayed a paroxysmal disruption of the background waveforms with an evolving ictal-like rhythm.**” We suspect that compared to source localisation, source reconstruction may require higher SNRs to resolve identifiable ictal features in source space. As well, certain geometries of anatomical structures, such as gyral areas, may impair MEG source reconstruction accuracy. These 11 seizures might be amenable to source reconstruction with corresponding ictal HDEEG signals (the subject of our future work in a larger cohort).”

The criteria we used to assess ViEEG results were similar criteria used to assess ictal behaviour with iEEG - the morphology and distribution of the signal, disruption of the background, and the nature of the early ictal rhythm field topography. At this stage, however, the quantitative nature of the relationship between our ViEEG and the iEEG remains unclear and is the subject of further investigation. Based on this proof of concept work, ViEEG-derived VIZ does have the potential to serve as a useful biomarker for the putative EZ. We have elaborated on this point (Line 322): “Ictal ViEEG signals from at least one seizure per patient present distinct characteristics of ictal events, such as hyper-synchronised rhythms, clear transitions from background activity to a seizure state, and spatial patterns of seizure propagation. Such qualitative characteristics of ictal ViEEG are also reflected by corresponding ictal iEEG data from the 6 patients whose iEEG data were available to us (Fig. 2, Supplementary Fig. 17). At this point, however, we lack a reliable quantitative measure to define the relationship between ViEEG reconstructed signals and the corresponding iEEG

discharges for morphology, spatial topography, and temporal evolution on an individual patient level. While requiring further investigation, this proof of concept work does, nonetheless, suggest that our ViEEG-derived VIZ does have the potential to serve as a useful biomarker for the patient's putative EZ."

4. Considerable Number of Removed Seizures - II. Following my previous comment, how did you determine the good quality of the seizures you kept? I believe you have the intra-cranial EEG recordings in 9 of these patients; given that typical seizures have more or less the same properties (given that iEEG and E/MEG might not have been recorded simultaneously), can you use objective measures such as correlation, spectral features, etc. to assess how well the estimated sources match/mismatch the underlying sources? Is there a more objective method to make this determination?

Response: As per the previous comment, we did not apply a more quantitative method to determine signal correlation between ViEEG and iEEG. Despite this, the results were encouraging for the majority of the seizures. It is important here to note that ViEEG was not expected to completely mimic the iEEG ictal characteristics. As per the Introduction, our question was: "Can we reconstruct ictal ViEEG signals that have distinct spatial and temporal characteristics of epileptiform discharges?" We did not expect the modelling to generate signals that were identical to the epileptiform signals detected by iEEG. This is because we are using a whole-head technique against a spatially constrained one in iEEG, and we are using a different recording modality in MEG. We state in the Discussion that "We have demonstrated that non-invasive ictal ViEEG signals preserve the most important characteristics for spatial distribution and morphology". On reflection, we agree this might mis-lead the reader to assume that ViEEG is a direct representation of the iEEG. We have amended this (Line 321): "We have demonstrated that non-invasive ictal ViEEG signals **contain meaningful temporo-spatial data to assist characterization of the putative EZ.**".

In Supplementary material (Line 62-66), we qualify this as well: "We visually inspected each ViEEG seizure and ensured all ViEEG seizures analysed by dynamical network models have 1) visible transition from background activity to ictal waveforms that is aligned in time with seizure onset annotated by C.P. using MEG sensor signals, 2) distinctive morphological features and spatial distributions of ictal waveforms that can resemble seizures recorded by iEEG, if iEEG is done.

5. Considerable Number of Removed Seizures - III. It seems that most of these discarded seizures (8 out of 11) are coming from patients 1 and 4. Are these all typical seizures in these patients? I mean from the 6 seizures recorded in patient 1, are they all the typical seizures observed in this patient? Or the 5 that “failed” are another type? Basically, have you looked more carefully into the reason these seizures “fail”? Is the SNR or recording quality the problem? Do they belong to another category compared to the seizures that worked?

Response: As per our comments for point 3, the “failed” seizures were of the same sub-type but they carried more noise contamination in the form of muscle artefact. And as per the above, we have added an example of a “failed” seizure to highlight these characteristics.

6. Epileptogenic Zone. The authors use the term EZ interchangeably with resection (if my understanding is correct). If all the patients studied here were seizure-free we could take the resection as the pseudo-EZ as the removal of those regions has stopped seizures, but the study includes non-seizure-free patients (which might be due to not fully treating potential EZ suggested by your algorithm). I think you might want to either caution about this in the manuscript or use another term such as “proposed EZ” or “clinically determined/proposed EZ”.

Response: Yes, we agree that the EZ is a theoretical construct and should not be used interchangeably with the resection volume, even in circumstances of a long term seizure free surgical outcome. With this in mind, we were careful to describe our work as an attempt to “characterise” the EZ, as per the concluding comments in the Introduction. We agree that there are other occasions when the term should be further qualified as either the “putative” EZ, the “proposed” EZ, the “likely” EZ, the “suspected” EZ, or the “notional” EZ and we have gone through the manuscript and made these changes as recommended.

Suggestion - 1. One of the shortcomings of beamforming methods, specifically older versions, is that correlated and coherent sources are not separated/estimated well, which is a potential issue in ictal imaging as you have discussed in the manuscript and the supplementary. However, I believe you can assess this with some simulations in your work. You can randomly assign some nodes of activity in the brain, assign some of them as the EZ, run the theta model with some connectivity matrix, run the forward problem to simulate MEG measurements at some frequency bands (add interference and noise at

source level and sensor level) and perform your proposed algorithm on this simulated data to assess how different source configuration, interference/noise level and its distance to EZ, effect of virtual electrode selection (during the inverse process), etc. affect your results specifically the beamforming portion of it. The whole framework in its entirety has not been checked yet (even though some of the parts have been separately tested in your previous publications).

Response: We agree that it would be helpful to assess the whole framework of our approach with simulation examples. We have now included examples in the Supplementary Material (Simulation Experiments: Supplementary Figs. 18-23).

Suggestion - 2. Is it possible to share your codes in this new framework, from A to Z, specifically with simulation data (that does not have the sensitivity of clinical data) to benefit the community?

Response: Yes, we are happy to share the codes for the simulation dataset that we have added. The code is available in the repository under the directory “supplemental_materials_ground_truth”

Minor Issues.1. ViEEG Numbers. Could you kindly include the number of ViEEG electrodes for each patient as well as the number of iEEG electrodes in some existing/new table? I know that you do not increase 500 based on the manuscript, but I was not sure if you fixed your electrode numbers to 500 or some number or not. I was thinking that if it is based on E/MSL size (plus resection margin), it will vary and since you are keeping the distance as 1 cm, the number will be different for each patient.

Response: Yes, we think this is an important point. We have now added a new Table in the Supplementary Material (Supplementary Table 1) to illustrate the interpatient variability for ViEEG sensor count relative to iEEG electrode count.

2. Brain View. The brain sub-figures containing clinical findings such as resection and SOZ, are different from the E/MSL figures, particularly the view. It is not easy to visually check the resection on top of your results. Is it possible to keep the view (brain orientation, view, etc.), the same for all sub-figures of all figure, so that it is easier to visually inspect the results? For instance, look at Fig. 3.

Response: Wherever possible, we have adjusted the images to try to improve the ease of comparison between sub-figures. Hence, we have made the following adjustments: **Fig. 3** (re-oriented VIZ results to common view), **Figs. 4, 5, Supp. Figs 7, 8** (sLORETA and resection images enlarged), **Supp. Figs 5, 9, 14, 16** (iEEG grid and resection images enlarged), **Supp. Figs. 6, 10, 11, 12** (resection image enlarged), **Supp. Fig. 13** (resection and VIZ images enlarged), **Supp. Fig. 15** (re-oriented and enlarged VIZ images, enlarged iEEG and resection).

3. E/MSL Threshold of 0.8. How was the threshold of 0.8 chosen for sLORETA results?

Response: We agree that this remains an arbitrary threshold in the source imaging literature. We use an 80% threshold to ensure the source maxima is reliably represented without the sLORETA probability map smearing to encompass lower probability solutions (Cosandier-Rimele et al, 2017). A larger area (lower threshold) distributed map is more likely to overlap with any resection boundary or with the iEEG results, leading to the possible overestimation of the accuracy of the sLORETA MEG source results. We have added a note on this point now to line 204 (Supplementary Fig. 3) “...in this paper we presented results from the top 20% threshold of VIZ nodes to be defined as hotspots to ensure that our work has the same thresholding strategy that was used by HDEEG and MEG source localisation in our previous publication¹⁰ **This threshold accommodates source localisation probability map sLORETA maxima without excessive smearing of the solution at lower thresholds (Cosandier-Rimele et al, 2017), which can lead to an overestimation of the accuracy of results based on the degree of overlap with the iEEG localisation and resection margins.**

Regarding point 1, can you change this threshold for ViEEG location selection so that you do not rely on surgical resection margin, at all?

Response: Yes, the threshold for ViEEG location can be altered to exclude the surgical resection bed. This is best tested with a prospective study design. We have added the following to the amendment above: “...localisation and resection margins. **Optimal thresholding for ViEEG requires further exploration in a prospective study when ‘virtual’ resection margins are defined before surgery.**

4. Typo in Supplementary. In page 2 of supplementary line 50 I believe it should read “... because removing it has the effect ...” which currently is “...removing it as the effect

Response: Thank-you for the comment. We have corrected the error.

...”.**Reviewer #2 (Remarks to the Author):**

This paper presents an original and interesting approach to finding the epileptogenic zone from the analysis of seizure data recorded with MEG. Using a network analysis and a virtual resection approach, the method provides a definition of the EZ. This definition is compared to the more traditionally defined SOZ (from MEG or EEG source modeling) and to the resection in 12 operated patients. Results indicate a solid predictive value for the EZ defined by the network approach. The paper is not easy to read in part because the method is complex. The paper cannot be read without frequent reference to the supplementary material, which is very extensive.

A significant weakness of the paper is the small number of patients. I realize that patients with seizures recorded during MEG are infrequent but this very fact reduces considerably the applicability of the method (see below).

Response: As per our comments to Reviewer 1, we agree that the study has a limited number of patients. The COVID pandemic has meant that our MEG facility has been in lockdown for over 18 months and hospital waiting times for standard inpatient epilepsy surgery work-up and elective operative lists have ballooned. Because we also need to follow up these patients for at least 12 months after their surgery to properly assess their outcome, the recruitment of new patients would potentially extend this work by another two years. We would stress that this was a proof of concept study as a prelude, perhaps, to a larger clinical study. Our patient cohort was quite heterogeneous for focal epilepsy sub-type and, to our knowledge, there are only three MEG studies that contain more seizures than our study. Ramanujam et al. (2017) described a cohort of 40 patients with at least one seizure but only 5 patients had undergone surgery (as opposed to all patients in our study). Medvedovsky et al. (2012) reported 47 patients, but only 11 patients gave a localizable result (as opposed to our study where all 12 patients gave a localisable result). Alkawadri et al (2018) reported 44 patients who experienced at least one seizure during the MEG recording but, as a retrospective study, surgical follow-up data was only available in 12 patients. The surgical follow-up in our 12 patients was based on prospectively acquired monthly post-operative seizure counts.”

We have therefore included these points in the section on Limitations (Lines 427-429).

“Our study is retrospective with a modest number of patients and seizures analysed compared to studies using iEEG seizures, although it contains one of the highest seizure counts obtained across ictal MEG studies. Our patient cohort was also quite heterogeneous for focal epilepsy sub-type. To our knowledge, to date there are three MEG studies that contain more seizures than our study. Ramanujam et al. (2017) described a cohort of 40 patients with at least one seizure but only five patients had undergone surgery (as opposed to all patients in our study). Medvedovsky et al. (2012) reported 47 patients, but only 11 patients gave a localizable source imaging result (as opposed to our study where all 12 patients gave a localizable result). Alkawadri et al (2018) reported 44 patients who experienced at least one seizure during the MEG recording but, as a retrospective study, surgical follow-up data was only available in 12 patients. The surgical follow-up in our 12 patients was based on prospectively acquired monthly, long-term post-operative seizure counts by Plummer et.al³⁶

There are only 6 seizure-free patients. Many researchers would have considered all patients with Engel class I outcome in one group, and I assume that given the size of this group, results would have been quite different then.

Response: We wanted to set the bar higher for post-operative outcome. The Engel classification, as a largely qualitative scale, is open to subjective interpretation. This is why we also quantified the outcome as a percentage seizure change in our original prospective study (Plummer et al, 2019). Unfortunately, attempts at this additional classification are rarely done in such epilepsy surgery studies (arguably because the majority of such studies are retrospective). With knowledge of the seizure percentage change, we were in a more assured position to state that patients with a 100% seizure reduction were seizure free, a better marker of EZ removal than Engel I alone, which allows for non-disabling seizures. We have clarified this point of separation by adding (Line 229): **We applied the combination of Engel score (as a largely qualitative measure) and the quantitative measure of percentage seizure change (as determined prospectively in our original study) (Plummer et al, 2019) to classify patients as genuinely seizure-free or not. We propose that this combined outcome measure is more discriminative when linking the resection bed to the notional EZ in seizure-free patients.**

I was surprised to see that all so-called virtual EEG locations were located on the outer mantle of the brain, in the same region as subdural electrodes are located. Source analysis is usually performed on the surface of the cortex, including in sulci. This is

particularly striking because MEG is especially sensitive to generators located in sulci and less so to generators located in gyri. The authors should justify this choice and should probably change their nomenclature: the virtual electrodes are in the location of subdural electrodes, mostly on the inner surface of the skull; the term “intracranial” is more general and I certainly had expected virtual electrodes within sulci of in the depth of the brain when reading the title and the abstract.

Response: Our virtual node placement did incorporate the sulcal depths, rather than just the sulcal rims and gyral crowns. We have added a new panel to Supplementary Figure 2 to illustrate this point (Line 118). “When defining ViEEG, locations of ViEEG electrodes are ensured to lie inside the inner-skull surface and predominantly in the cortical grey matter involving both gyral and sulcal surfaces. An example of ViEEG electrode locations is demonstrated here (Supplementary Fig 2C) with brain model (transparency 50%) and MR images. The red-coloured ViEEG electrode in the brain model figure (Supplementary Fig 2C left panel) is also coloured in red in the MR images (Supplementary Fig 2C right panel)”.

The patient selection process must be described in detail to be able to judge the generalizability of the results. It is unusual to record seizures during a 1-hour MEG recordings and 12 of the 13 studied patients had seizures, in fact quite a few seizures. Among how many consecutive MEG recordings were these patients selected and on what basis? In my experience, it is one every 15 or 20 patients who has a seizure during a MEG study. The general pool of patients from which these were selected must be described.

Response: Yes, we agree this is potentially misleading. We were actually referring to the 13 patients described in the original publication (Plummer et. al, 2019), 12 of whom had at least one seizure in the MEG. The 12 patients were part of 50 consecutively recorded patients with drug resistant epilepsy who had daily to weekly seizures and frequent discharges on routine pre-surgical video EEG telemetry. Our seizure rate is higher than usual at around 25% of studies for several reasons: we sleep deprive (to 4 hours) the evening before the recording, withhold medication for 12 hours before the recording, and recruit surgical candidates with the most severe epilepsy as we have access to the only MEG scanner in Australia that is set-up to do this work. From 50 consecutively recorded epilepsy surgery candidates, twelve patients had at least one seizure in the MEG scanner (seven males, six females, age range 10-54 years, median 29 years; disease duration 3-32 years). A total of 36 seizures were captured

from this cohort who had severe drug-resistant focal epilepsy (daily to weekly seizures and frequent discharges on routine video-EEG telemetry) with either no visible MRI lesion (9 patients) or a complex lesion (3 patients). All patients had at least 20 months post-surgical follow up (median 24 months, range 20-39 months). (Line 113 – 119).

We give reasons for the higher than expected seizure rate and we indicate our aim to apply this analysis to interictal data as well in the section on limitations and future directions (Line 454-461) And while we accept that our seizure rate in the MEG is higher than usual at around 25% of studies performed (12/50 patients), there were several possible explanations. We sleep deprive patients and withhold medication for 12 hours before the recording, and we recruit surgical candidates at the more severe end of the seizure frequency spectrum based on the clinical history and the frequency of interictal and ictal events during routine video-EEG telemetry. Nonetheless, given the challenges faced in capturing seizures during a one-hour MEG recording, the next step is to use interictal data from a larger patient cohort to potentially broaden the clinical utility of our method.

The discussion must include a clear statement regarding how often this method is applicable in a group of patients who are candidate for epilepsy surgery.

Response: Yes, we agree this is an important point to make. We have added the following points to the end of the Discussion (as above): **And while we accept that our seizure rate in the MEG is higher than usual at around 25% of studies performed (12/50 patients), there were several possible explanations. We restrict sleep to four hours on the eve of the recording, we withhold medication for 12 hours before the recording, and we recruit surgical candidates at the more severe end of the seizure frequency spectrum based on the clinical history and the frequency of interictal and ictal events during routine video-EEG telemetry.**

Two apparently contradictory statements are given: “ViEEG was defined for each patient solely using information from MSL31 (not ESL)” and “M.C. had no knowledge of which solution (early-ESL or early-MSL) was the earliest solution for each patient when defining ViEEG”. This needs an explanation.

Response: Yes, as per the similar comment from Reviewer 1, we have clarified this statement as follows. While M.C. was aware of the locations of the MEG Source Localisation (MSL) given by sLORETA, it is important to note that each MSL solution had 3 sub-solutions based on the time-point used for sLORETA modelling of the averaged ictal

discharge – early, mid, and late (Plummer et al, 2019). ‘Early’ was the earliest latency sLORETA MSL solution that reached 90% of the measured signal variance from discharge take-off; ‘Mid’ was the MSL solution at the half-way time-point between take-off and the averaged ictal discharge peak; ‘Late’ was the MSL solution at the time-point of the peak. Plummer et al. (2019) found that the early solution was a better predictor of the epileptogenic zone (EZ) than the mid and late solutions and that the locations of the 3 solutions were typically quite dispersed (arguably from cortico-cortical propagation). While author M.C. placed the ViEEG to cover all 3 solution locations, M.C. was blinded to the phase information (that is, M.C. did not know which of the 3 solutions was the early solution). Please also note that M.C. was blinded to and did not use the corresponding ESL solutions to guide the placement of the ViEEG. Indeed, Plummer et al. (2019) found that the early-ESL solutions were a better predictor of the EZ than corresponding early-MSL solutions for ictal discharges, while early-MSL was superior to early-ESL for interictal discharges.

We have amended Line 163-168: ViEEG was, **therefore**, defined for each patient using information from MSL³⁶ **and not ESL in order to limit any subjectivity tied to the manual selection of ViEEG locations**. Thus, the locations of ViEEG electrodes extensively covered MSL of averaged ictal discharges (early-, mid-upstroke and late-peak phase solutions)³⁶. **M.C. did not know the phase order of the MSL solutions, nor the solution modality (ESL or MSL) that gave the best predictor of the proposed EZ in a given patient.**

We have also amended line 152 to avoid confusion as this is a MEG only analysis: We propose a novel concept (ViEEG) that consists of multiple virtual electrodes or virtual sensors guided by MEG source imaging^{35, 37}.

I am confused by the statement on lines 228 and following: the authors write that the real iEEG and the virtual iEEG signals have a similar distribution. I find them very different in their morphology and spatial distribution. I am not sure how to compare locations though. Unfortunately the supplementary material does not show real iEEG seizures to allow comparison in other cases. Making such a statement and providing only this one non-convincing picture is not appropriate. The authors talk about virtual iEEG and there should be information on how the virtual EEG resembles the actual iEEG. Given the apparent lack of resemblance and the absence of demonstration of this resemblance, it is more appropriate to talk about a method to predict the EZ, but not about virtual EEG.

Response: Yes, as also raised by Reviewer 1, we agree that the relationship between iEEG and ViEEG signal features needs to be more clearly described. As per the comment to Reviewer 1, the criteria we used to assess ViEEG results were similar criteria used to assess ictal discharges on iEEG - the morphology and distribution of the signal, disruption of the background, and the nature of the early ictal rhythm spatial topography. We have now added an image of the iEEG rhythm at seizure onset for the other 5 patients whose iEEG traces were available (Patients 1, 4, 10, 11, 12) (Supplementary Fig. 17). At this stage, however, the quantitative nature of the relationship between our ViEEG and the iEEG remains unclear and is the subject of further investigation. Based on this proof of concept work, ViEEG-derived VIZ does have the potential to serve as a useful biomarker for the putative EZ. We have elaborated on this point (Line 322): “Ictal ViEEG signals from at least one seizure per patient present distinct characteristics of ictal events, such as hyper-synchronised rhythms, clear transitions from background activity to a seizure state, and spatial patterns of seizure propagation. Such **qualitative** characteristics of ictal ViEEG are also **reflected** by **corresponding** ictal iEEG data from the 6 patients **whose iEEG data were available to us** (Fig. 2, **Supplementary Fig. 17**). **At this point, however, we lack a reliable quantitative measure to define the relationship between ViEEG reconstructed signals and the corresponding iEEG discharges for morphology, spatial topography, and temporal evolution on an individual patient level. While requiring further investigation, this proof of concept work does, nonetheless, suggest that our ViEEG-derived VIZ does have the potential to serve as a useful biomarker for the patient’s putative EZ.**

To avoid overstating the findings, as previously mentioned, we have amended Line 312 : We have demonstrated that non-invasive ictal ViEEG signals contain meaningful temporo-spatial data to assist characterisation of the putative EZ. In the supplementary material (Line 62), we qualify this as well: “We visually inspected each ViEEG seizure and ensured all ViEEG seizures analysed by dynamical network models have 1) visible transition from background activity to ictal waveforms that is aligned in time with seizure onset annotated by C.P. using MEG sensor signals, 2) distinctive temporal features and spatial distributions of ictal waveforms that can **resemble** seizures recorded by iEEG, if iEEG is done.

Figure 3. Such a case does not support a more extended SOZ. It only states that the resected region is not the epileptogenic zone, which could be small but located in a different region. This is not known. In the figures, the resection should be shown on the

same map as the hotspot of the VIZ; otherwise they are difficult to compare. The boundaries of the VIZ do not seem very meaningful and could be ignored if the figure becomes too complex.

Response: We agree that the likely EZ in this case is unclear. We have amended the comment (Line 807): “. It is possible that the **concordant VIZ hotspots and the iEEG flag a more extensive EZ or a separate EZ network.**” We have amended some of the Figures and done our best to juxtapose the resection with all the localisation results. We think the Boundaries are important to show as well to contextualise the hotspot results.

The F-score is used to report the results in the main part of the manuscript but this is not easy to interpret, particularly as it incorporates the VIZ boundary, which does not seem to be very meaningful. I think it would be more informative to have the precision and recall results, which are more practically interpretable (now in supplementary material).

Response: Again, we agree that this is a good point. We have therefore switched Figure 6B in the Main paper with Supplementary Fig. 3 and amended the text in the Results accordingly. Figure 6B shows the precision, or positive predictive value, of the VIZ hotspot (top 20% VIZ nodes ranked by NI) and the recall, or sensitivity, of the VIZ boundary in predicting the resection margin and the earliest solution (Line 845). **B) Precision (or positive predictive value) and recall (or sensitivity) for AEC-VIZ and MI-VIZ in predicting the resection margin (top panel) and the earliest solution (bottom panel) are presented in boxplots (horizontal bar, box upper boundary, box lower boundary and dots represent median, first and third quartile and each individual VIZ, respectively). This shows that the MI-VIZ recall (for VIZ boundary) sufficiently captures the entirety of resection margin and the earliest solution and identifies non-ictogenic brain areas that are less likely to overlap with the EZ and are therefore potentially less concerning for iEEG coverage. Moderate precision values (for VIZ hotspot) are found for both AEC-VIZ and MI-VIZ in predicting the resection margin and earliest solution. MI-VIZ hotspots appear to have higher precision than AEC-VIZ hotspots in predicting the resection margin and, to a lesser degree, the earliest solution. The spatial overlap between VIZ and clinical localisation are demonstrated on a per patient and per seizure basis in Supplementary Table 4.**

We have referred to the Figure in the Discussion (Line 353). “This study demonstrates proof-of-concept that dynamical network models using ViEEG signals identify a sub-network VIZ that provide a valid characterisation of the EZ and prediction of the clinical localisation (Fig. 6, Supplementary Fig. 4)”

The authors conclude that the non-linearly defined networks give better results than the linearly defined networks, and make other such comparisons. Given the small number of patients and the large scatter of the results, I do not think it is appropriate to make such conclusions when there are no statistical comparisons between the measures.

Response: Yes, we agree that we may be overstating the significance of our results here given the relatively low number of patients. Our results demonstrate that MI-VIZ and AEC-VIZ **assist** characterising the EZ and predicting the clinical localisation (Fig. 6, Supplementary Fig. 4), which suggests a connectivity approach that captures both linear and non-linear interactions might offer more information about ictal network structures than an approach that only captures **one type of** interaction. This finding also lends support to a previous theoretical model using nonlinear dynamics from the seizure onset zone to identify the EZ from seizure propagation and predict seizure propagation and termination^{16, 64}..

We have also amended the subtitle in the Results: **MI-VIZ may predict the putative EZ and clinical localisation better than AEC-VIZ**

The authors write “This lends confidence to our approach, as consistency of VIZ hotspots was found between seizures for a given patient – refer to Patient 2 (Supplementary Fig. 6), Patient 8 (Supplementary Fig. 12), and Patient 11 (Supplementary Fig. 15)”. There is no quantification of the overlap from seizure to seizure and there are patients for whom there is very poor overlap. The authors should either remove this statement or also include the cases of discordance and discuss them.

Response: Yes, we agree and we have removed this statement now (Line 363-365):

On the subject of multiple seizures, this may be explained somewhere, but it is not clear to me how the study dealt with the several seizures of one patient when making the patient-based predictions.

Response: Yes, we agree that it is important to provide a strategy of making patient-based predictions given solutions derived from ViEEG and dynamical modelling of multiple seizures. In this study, we treated each seizure independently and hence, we did not average or choose a seizure out of multiple ones recorded by MEG to represent the solution for a patient (i.e., the patient-based prediction). One of the key findings from this study is both consistency and variability of VIZ (AEC-VIZ and MI-VIZ) solutions derived from multiple seizures have been observed in one patient. Providing a strategy that incorporates VIZ solutions from multiple seizures is not the focus of this study; this study aimed to answer the question whether the dynamical models developed using invasive electrophysiology can be translated to non-invasive source space. The future work is to assess and devise a strategy of incorporating solutions from multiple seizures and multiple dynamical models for a patient, which is then presented to the treating clinicians for prospective study or clinical trial.

Reviewer #3 (Remarks to the Author):

The authors propose a combination of MEG source localization and dynamic network modeling to create virtual intracranial EEG (ViEEG) solutions at the cortical level that can estimate virtual ictogenic zones (VIZs) from non-invasive ictal MEG recordings. They consider two measures of connectivity to be used for simulating network activity using the theta model, namely Pearson correlations of the envelopes and mutual information. They comprehensively validate the estimated VIZ hotspots and boundaries, and demonstrate high precision/recall with respect to the clinically detected epileptogenic zones. This is quite a well-written manuscript, the results are impressive, and the methods are thoroughly thought out, applied, and carefully explained. In particular, the results serve as a stepping stone for several future studies enabling further clinical validations and using both MEG and EEG to improve ViEEG estimation.

Response: Thank-you for the comments.

I have the following comments for the authors:

1. While references 23 and 24 (lines 79 and 81) are relevant to the description of EEG and MEG and their shortcomings, I suggest that the authors add a few more standard

references and/or surveys for EEG and MEG source localization as well. In particular, 23 is more relevant to magnetic shielding (than the nature of EEG and MEG signals and their relation to cortical activity). Some suggestions: Hamalainen et al. (1993), Baillet et al. (2001), Schoffelen & Gross (2009), Hari & Puce (2017), etc. 2.

Response: We agree and have added citations now.

The authors give a compelling argument in the discussion that using MEG (instead of EEG) is preferable due to less sensitivity to the type of conductance-based head model used, but all along I was wondering why the LCMV was not jointly applied to MEG + EEG data. I suggest adding a justification to this effect in the methods section Virtual iEEG (lines 153-154).

Response: Yes, this needs to be clarified. As per the reply to Reviewer 1 (point 1), because this study has only looked at the MEG ictal data, we have removed the term ‘HDEEG’ from line 149: “We propose a novel concept (ViEEG) that consists of multiple virtual electrodes or virtual sensors guided by MEG source imaging^{30, 32} “ On the important point of using the simultaneously recorded HDEEG as well, we would stress that, as a proof-of-concept study, we wanted to simplify the analysis to one modality. We chose MEG because, to our knowledge, ictal MEG has not been subjected to this kind of analysis before in a clinical population and because the signal is relatively unaffected by tissue planes. Also, as per Plummer et al. (2019), there are often lead-lag differences between HDEEG and MEG for relative timing of ictal onsets. This added complexity of combining the signals in the same source space was a possible explanation given by Plummer et al (2019) for the inferior performance of combined HDEEG-MEG source localisation against independent HDEEG and MEG source localisation. In the present study, we did use the HDEEG to confirm an ictal rhythm in the MEG recording. We have expanded on these points now (Line 153-157). **While our original source imaging study³⁶ used HDEEG and MEG data, the present study only uses MEG data. This is because of the added complexity of combining HDEEG-MEG signals in the same source space given the relative differences between modalities for ictal onset latencies and tissue conductivity effects. Here, the simultaneously acquired HDEEG was used to help confirm an ictal MEG rhythm.** An example of **MEG-informed ViEEG** for a patient is given in Fig. 1

3. While the explanation of the Blinded Analysis is worthwhile, as it suggests that possible subjective biases have been eliminated from the analysis, it may be counterproductive to more complex follow-up studies. The full knowledge of the data and clinical attributes by all the team members can help optimize the choice of the parameters, models, etc. The subjective biases in this case can be avoided by the common separation of the data into training and testing sets. I suggest that the authors give a more clear justification of why such blinded analysis was necessary.

Response: We wanted to minimise bias that is more likely to arise in a retrospective study such as this one. While large parts of the analysis are automated, a potential source of bias comes with the manual placement of the ViEEG virtual electrodes across the brain compartment. We were concerned that prior knowledge of the most accurate localisation result given by HDEEG or MEG source imaging in the original prospective study (Plummer et al) would introduce bias in the ViEEG sensor location selections. While we agree that the availability of additional clinical information might help fine tune the analysis, we did not want this proof of concept study to be contaminated by concerns of introduced bias. We have clarified this point with the added comment (Line 163): “, ViEEG was, therefore, defined for each patient using information from MSL and not ESL in order to limit any subjectivity tied to the manual selection of ViEEG locations

4. In Fig. 2, the iEEG traces are much more spatiotemporally localized than the ViEEG traces, i.e., the temporal patterns of the hippocampal sources are quite distinct from the basal and lateral sources. This is not the case for ViEEG traces, for which the temporal patterns seem more correlated across the cortical areas. Is this a result of the source mixing due to poor localization? Please clarify.

Response: Yes, as also raised by the other Reviewers, we agree that the relationship between iEEG and ViEEG signal features needs to be more clearly elaborated. We agree that the apparent ViEEG ictal waveforms, such as those shown in Fig. 2 and Supplementary Fig. 17, are not identical to the corresponding iEEG ictal waveforms. We did not expect the waveforms to mirror each other but instead hypothesised that the ViEEG reconstructed ictal signals could serve as a useful biomarker of the possible EZ. As per the previous comments to Reviewer 1 and Reviewer 2, the criteria we used to assess ViEEG results were similar criteria used to assess ictal behaviour with iEEG - the morphology and distribution of the signal, disruption of the background, and the nature of the early ictal rhythm spatial

topography. At this stage, however, a more direct quantitative characterisation of the relationship between our ViEEG and the iEEG remains unclear and is the subject of further investigation. Based on this proof of concept work, ViEEG-derived VIZ does have the potential to serve as a useful biomarker for the putative EZ. We have elaborated on this point (Line 322): “Ictal ViEEG signals from at least one seizure per patient present distinct characteristics of ictal events, such as hyper-synchronised rhythms, clear transitions from background activity to a seizure state, and spatial patterns of seizure propagation. Such **qualitative** characteristics of ictal ViEEG are also **reflected** by corresponding ictal iEEG data from the 6 patients whose iEEG data were available to us (Fig. 2, Supplementary Fig. 17). **At this point, however, we lack a reliable quantitative measure to define the relationship between ViEEG reconstructed signals and the corresponding iEEG discharges for morphology, spatial topography, and temporal evolution on an individual patient level. While requiring further investigation, this proof of concept work does, nonetheless, suggest that our ViEEG-derived VIZ does have the potential to serve as a useful biomarker for the patient’s putative EZ.**” We have also amended the statement in Line 311: “We have demonstrated that non-invasive ictal ViEEG signals contain **meaningful temporo-spatial data to assist characterization of the putative EZ.**” . In Supplementary material (Line 62), we qualify this as well: “We visually inspected each ViEEG seizure and ensured all ViEEG seizures analysed by dynamical network models have 1) visible transition from background activity to ictal waveforms that is aligned in time with seizure onset annotated by C.P. using MEG sensor signals, 2) distinctive morphological features and spatial distribution of ictal waveforms that can **resemble** seizures recorded by iEEG, if iEEG is done (Fig. 2, Supplementary Fig. 17). For clarity we have also amended the Figure Legend for Fig. 2 (Line 786): “**While carrying a slightly different morphology to the corresponding iEEG ictal rhythm,** distinct ictal waveforms are seen in the left anterior hippocampal structure and left basal temporal region from a ViEEG seizure aligned in time with seizure onset identified by MEG sensor signals.”

5. The examples referred to on lines 277-278 are a bit unclear. It seems that Patient 6 is an example of AEC-VIZ being a better predictor of iEEG SOZ, and Patient 12 seizure 1 is an example of AEC-VIZ being a better predictor of mid-MSL. But it is stated that both are examples of MI-VIZ hotspots being better predictors of the resection margin and the earliest solution. While the latter is the case for Patient 12, it doesn't seem to hold for Patient 6. Please clarify.

Response: Yes, we agree that this is confusing on a few counts – iEEG was not done for Patient 6 and this was a case where both the MI-VIZ and the AEC-VIZ hotspots predict the earliest solution and the resection margin but we wanted to stress that the AEC-VIZ map was less predictive of the putative EZ as it also included the failed first surgical resection bed, while the MI-VIZ map, with a higher NI value, only included the successful second surgery resection bed. We have clarified this at several points in the manuscript as follows.

Line 812, Fig. 4: “ Despite ViEEG being defined by only MSL solutions (only MEG data was used to reconstruct ViEEG signals in this study), dynamical network models suggest AEC-VIZ and MI-VIZ hotspots are concordant with the earliest source localisation solution, which was early-ESL in this case (not early-MSL), and the second resection (magenta). **However, note the more dispersed AEC-VIZ NI map that encompasses the first failed resection as well.**

Supplementary File, Line 395, Supplementary . Fig. 10. “**Compared to the AEC-VIZ hotspot, the MI-VIZ hotspot is, however, more localised to the early ESL solution and to the successful repeat surgery resection (while the AEC-VIZ also includes the failed first resection). Nonetheless, AEC-VIZ and MI-VIZ from MEG data better concords with the earliest solution given by the EEG rather than the corresponding MEG sLORETA solution.**”

Line 304: “Examples can be seen in Patient 6 (Fig. 4, Supplementary Fig. 10), **where the MI-VIZ hotspot predicts the successful repeat surgery resection bed while the more diffuse AEC-VIZ hotspot encompasses the first failed resection bed as well**, and in Patient 12 seizure 1 (Fig. 5, Supplementary Fig. 16), where **the MI-VIZ hotspot predicts the resection margin and the earliest solution, while the AEC-VIZ hotspot does not.**”

Also, the supplementary figures are mis-numbered: please change Supp. Fig. 10 to 9, and Supp. Fig. 16 to 15.

Response: Thank-you, but as we have added a new Supp Fig. the error corrects itself.

6. Supplementary material, line 11: please add a reference for the amplitude adjusted Fourier transform surrogate generation.

Response: Thank-you. We have added the citation here.

7. Supplementary material, line 28: The first term on the right hand side of the differential equation for θ_j must be $(1-\cos(\theta_j))$. The symbol θ_j is dropped. Please revise.

Response: Thank-you. We have amended the equation.

8. Supplementary material, lines 37-41: How is the "seizure state" defined for simulated activity using the theta model? Is it based on thresholding the amplitude of each node? Please explain.

Response: We have amended the text as follows (Line 39): “As in Goodfellow et al.¹, we quantified the dynamics of the system using the notion of brain network ictogenicity BNI , which is the average fraction of time that each node spends in the ‘seizure state’. To identify the seizure state, we transform the variable θ_i for node i using the function $T(\theta_i) = 0.5(1 - \cos(\theta_i - \theta_i^S))$ which takes values in $[0, 1]$ where a value near 1 indicates a spike. Then, for this node, we identify points where... $T > 0.9$. This marks the beginning of a node entering the seizure state. The node exits the seizure state if $T < 0.9$ for at least 24 time units ($\frac{24}{dt}$ time steps of the model), which indicates that no spikes have occurred. The BNI value is therefore obtained by computing the dynamical system over a long period of time (4×10^6 timesteps), with multiple runs to mitigate the effects of noise (128 noise runs) and averaging the time spent in the seizure state over all nodes, times, and runs.”

9. Supplementary material, lines 184-200: the AIC and BIC are reported, but it is not clear which criterion was used to determine the model order in the logistic regression. Please clarify.

Response: We have added the following by way of explanation (Line 220) Both BIC and AIC were used to compare AEC and MI methods for the logistic regression.

10. Line 424: Please change "ViEEG with on feasible network scale" to "ViEEG with a feasible network scale"

Response: Thank-you. We have amended this typo.

REVIEWER COMMENTS

Reviewer #1 (Remarks to the Author):

The authors have satisfactorily and thoroughly responded to my questions and concerns and I have no further comments.

Reviewer #2 (Remarks to the Author):

The authors have addressed appropriately most of my comments (reviewer #2). I remain with one significant problem and one of lesser importance and simple to address.

In response to my comment starting with "I was surprised to see that all so-called virtual EEG locations were located on the outer mantle of the brain", the authors show one example in a supplementary figure where the virtual electrode is presumably in a sulcus. I have to say that the right panel of supplementary figure 2C, even if I enlarge it, does not allow me to see an intracerebral electrode; there may be a bit of faint red coloring... Also, I do not understand: is the red electrode appearing on the dural surface in Supp 2C (left panel) actually in the depth but shown on the surface? This is confusing and should be clarified for all figures. Importantly, the authors do not address the broader question I raised: in all the examples shown in the main part of the paper the virtual electrodes appear to be on the dural surface. Virtual electrodes in a sulcus therefore appear to be very rare. Why would that be, given the higher sensitivity of MEG to sulcal generators? Are the electrodes shown on the dural surface actually on the dural surface? This question needs to be addressed with a substantial comment in the main part of the paper and throughout the paper if necessary.

In response to my last comment, starting with "On the subject of multiple seizures", I think the authors should mention in the limitations of the study that they have studied one seizure at a time, have not established the similarity between seizures nor how to integrate results from multiple seizures to come to a patient-based prediction.

Reviewer #3 (Remarks to the Author):

Thank you for addressing my previous comments thoroughly and constructively. I have no further comments.

RESPONSE TO REVIEWER COMMENTS

In order to distinguish the modified text from the earlier Revision 1 changes, we have underlined the Revision 2 changes.

Reviewer #1 (Remarks to the Author):

The authors have satisfactorily and thoroughly responded to my questions and concerns and I have no further comments.

Response: Thank-you Reviewer #1 for the comments.

Reviewer #2 (Remarks to the Author):

The authors have addressed appropriately most of my comments (reviewer #2). I remain with one significant problem and one of lesser importance and simple to address. In response to my comment starting with "I was surprised to see that all so-called virtual EEG locations were located on the outer mantle of the brain", the authors show one example in a supplementary figure where the virtual electrode is presumably in a sulcus. I have to say that the right panel of supplementary figure 2C, even if I enlarge it, does not allow me to see an intracerebral electrode; there may be a bit of faint red coloring

We have enlarged this image now to more easily show the virtual electrode in a sulcus at the base of the temporal lobe. The use of the red marker was to help orientate the MRI with the cortical reconstruction for the reader. With the aid of the corresponding MRI in this figure, note that the majority of the ViEEG electrodes are in fact within the cortical ribbon with most taking in part of the sulcal anatomy. Because we wanted to mimic the ICEEG 10 x 10 mm regular array with ViEEG 10 x 10 mm placements, there were always going to be some ViEEG electrodes that found a position within the mid-point of the gyral crown. To be clear though, the anatomical extent of ictal generators detectable by MEG is not in the order of mm but several cm (3-4 cm²)¹; thus, with inclusion of multiple peri-sulcal ViEEG positions within the virtual array, we were able to demonstrate the value of this approach in estimating the putative epileptogenic zone non-invasively. We agree that a MEG study that limits the ViEEG positions to sulci only might generate more accurate results. We are in a position to

do this now with the advent at our centre of sEEG, which is able to sample sulci more readily as opposed to the grids and strips that were used for the bulk of these cases where ICEEG sampling is limited to activity at the gyral crowns and superficial sulcal crests.

We have included this explanation in the Methods and Discussion now.

Methods Line 160: All cortical reconstructions are rendered 50% transparent to permit visualisation of the ViEEG electrodes so that they are actually deeper than they appear (they sit within the bed of the cortical ribbon and not superficially on the dural surface). Given the higher sensitivity of MEG to sulci and fissures, ViEEG electrode arrays were positioned to optimise coverage of the deeper peri-sulcal cortical ribbon and fissural surfaces within the constraints of this 10 mm x 10 mm electrode lattice

Discussion Line 475: Because we wanted to mimic the ICEEG 10 x 10 mm regular array with directly comparable ViEEG 10 x 10 mm placements, there were always going to be some ViEEG electrodes that found a position within the mid-point of the gyral crown (where MEG is unlikely to see the ictal discharge). To be clear though, the anatomical extent of ictal generators detectable by MEG is not in the order of millimetres but several centimetres (3-4 cm²)¹; thus, with inclusion of multiple peri-sulcal ViEEG positions within the virtual array, we were able to demonstrate the value of this approach for non-invasive estimation of the putative EZ. MEG signal analysis that limits the ViEEG positions to sulcal and fissural surfaces only may well generate more accurate results. We are in a position to do this now with the recent availability at our centre of sEEG, which is able to sample sulci more readily as opposed to the grids and strips that were used for the bulk of these cases where ICEEG sampling is limited to ictal activity at the gyral crowns and superficial sulcal crests

Also, I do not understand: is the red electrode appearing on the dural surface in Supp 2C (left panel) actually in the depth but shown on the surface? This is confusing and should be clarified for all figures.

As noted above, we agree we ought to clarify this point. The red electrode appears to be on the surface in this 2D image but please note that the cortical surface has been made 50% transparent to allow electrode positions to be seen more easily. The downside is that the electrode appears to sit on the surface but this is not the case as the ViEEG electrodes lie within the substance of the cortical ribbon, taking in the edge or wall of the sulcus (at depth) as much as can be allowed within the confines of a 10 x 10 mm configuration. This case is

also the only case that incorporated sEEG electrodes (in the form of bilateral hippocampal depth) where we had access to the ICEEG data. These electrodes have now been distinguished from the left lateral temporal grid electrodes by the placement of a hat symbol next to the hippocampal depth electrodes.

Supp Material Line 125: Because all cortical reconstructions are rendered 50% transparent to permit visualisation of the ViEEG electrodes, they are actually deeper than they appear (they sit within the bed of the cortical ribbon and not superficially on the dural surface)

Importantly, the authors do not address the broader question I raised: in all the examples shown in the main part of the paper the virtual electrodes appear to be on the dural surface. Virtual electrodes in a sulcus therefore appear to be very rare. Why would that be, given the higher sensitivity of MEG to sulcal generators? Are the electrodes shown on the dural surface actually on the dural surface? This question needs to be addressed with a substantial comment in the main part of the paper and throughout the paper if necessary.

As per the comments above, many electrodes are indeed sampling a portion of the sulcal wall at depth but the 10 x 10 mm grid will inevitably include ViEEG electrode positions that are sub-jacent to the gyral crown. The electrodes are not placed on the dural surface per se (but rather within the cortical ribbon), although the cortical reconstructions do give this appearance. As above we have expanded on this point now.

In response to my last comment, starting with “On the subject of multiple seizures”, I think the authors should mention in the limitations of the study that they have studied one seizure at a time, have not established the similarity between seizures nor how to integrate results from multiple seizures to come to a patient-based prediction.

Yes, we agree we should include this comment and we now include the following:

Discussion Line 498: It is important to note that only one seizure has been analyzed at a time, and that multiple seizures (within the same patient), although demonstrated here to give different results, have not been formally compared. An interesting extension to this work would be to compare different seizures and integrate these results into a patient-specific prediction of the EZ.

Thank-you Reviewer 2 for the helpful comments.

Reviewer #3 (Remarks to the Author):

Thank you for addressing my previous comments thoroughly and constructively. I have no further comments.

Thank-you Reviewer 3 for the comments

1. Ebersole JS, Ebersole SM. Combining MEG and EEG Source Modeling in Epilepsy Evaluations. *Journal of Clinical Neurophysiology* **27**, 360-371 (2010).

REVIEWER COMMENTS

Reviewer #2 (Remarks to the Author):

Unfortunately it seems that we have a misunderstanding: I stated that I was surprised that virtual electrodes were always on the surface of gyri and the authors responded by saying that the electrodes are in fact within the cortex, probably 1.5mm below the surface. My question was regarding the absence of virtual electrodes within sulci (2 or 3 or 4 cm below the pial surface of the brain as seen from a subdural grid, i.e. within the depth of a sulcus, independently of whether it is 1mm below the cortical surface of the sulcal wall). From the authors' answer, I have the feeling that virtual sensors were constrained to be on the outer surface of the brain, to mimic subdural electrode arrays. If this was the case, it must be clearly stated and thoroughly discussed, as this MEG constraint will necessarily increase the correlation between virtual and subdural electrode findings, compared to a situation where MEG sources could be anywhere.

Reviewer 2: Unfortunately it seems that we have a misunderstanding: I stated that I was surprised that virtual electrodes were always on the surface of gyri and the authors responded by saying that the electrodes are in fact within the cortex, probably 1.5mm below the surface.

My question was regarding the absence of virtual electrodes within sulci (2 or 3 or 4 cm below the pial surface of the brain as seen from a subdural grid, i.e. within the depth of a sulcus, independently of whether it is 1mm below the cortical surface of the sulcal wall). From the authors' answer, I have the feeling that virtual sensors were constrained to be on the outer surface of the brain, to mimic subdural electrode arrays.

If this was the case, it must be clearly stated and thoroughly discussed, as this MEG constraint will necessarily increase the correlation between virtual and subdural electrode findings, compared to a situation where MEG sources could be anywhere.

Response: Thank you for allowing us to clarify this point, which we hope we have now satisfactorily addressed below.

Methods Line 160: All cortical reconstructions are rendered 50% transparent to permit visualisation of the ViEEG electrodes so that they are actually deeper than they appear (they sit within the bed of the cortical ribbon and not superficially on the dural surface). Given the higher sensitivity of MEG to sulci and fissures, ViEEG electrode arrays were positioned to optimise coverage of the deeper peri-sulcal cortical ribbon and fissural surfaces within the constraints of this array with 10 mm x 10 mm inter-electrode distances. We configured the ViEEG array in this manner to mimic the array given by the subdural grid to allow a more direct comparison of ictal waveforms between actual and virtual intracranial signals across comparable sensor geometries.

Discussion Line 475: Because we wanted to mimic the ICEEG 10 x 10 mm regular array with directly comparable ViEEG 10 x 10 mm placements, there were always going to be some ViEEG electrodes that found a position within the mid-point of the gyral crown (where MEG is unlikely to see the ictal discharge). We accept that the choice of a more restricted ViEEG alignment to the outer cortical surface, with incomplete representation of deeper sulcal clefts, reduces the sensitivity of the MEG signals guiding the ViEEG reconstruction. Nonetheless, in spite of the constraint of a subdural grid-like ViEEG configuration that was

not ideally suited to detection of tangential sulcal sources, it is encouraging that our ViEEG results were clinically informative. We also recognise that this constraint may also increase a correlation between superficial virtual and subdural electrode findings compared to a situation where ViEEG sources are positioned deeper to the cortical surface. However, because our surgical cohort was primarily investigated with subdural grids at the time, our ViEEG results had to be tested against this reference gold standard to satisfy our proof-of-concept study design. We also migrated ViEEG grids to encompass a larger total cortical surface area than covered by the fixed subdural grid locations and, despite this, our VIZ results met statistical significance for predicting the likely EZ. To be clear though, the anatomical extent of ictal generators detectable by MEG is not in the order of millimetres but several centimetres (3-4 cm²)¹; thus, with inclusion of multiple peri-sulcal ViEEG positions within the virtual array, we were able to demonstrate the value of this approach for non-invasive estimation of the putative EZ. MEG signal analysis that limits the ViEEG positions to **just** sulcal and fissural surfaces may well generate more accurate results. We are in a position to do this now with the recent availability at our centre of stereoEEG, which is able to sample sulci more readily as opposed to the grids and strips that were used for the bulk of these cases where ICEEG sampling is limited to ictal activity at the gyral crowns and superficial sulcal crests